

# Retrieval and evaluation of tropospheric aerosol extinction profiles using MAX-DOAS measurements over Athens, Greece

Myrto Gratsea[1,2], Tim Bösch[3], Panos Kokkalis[4,8], Andreas Richter[3], Mihalis Vrekoussis[5,6], Stelios Kazadzis[7,1], Alexandra Tsekeri[4], Alexandros Papayannis[9], Maria Mylonaki[9], Vassilis Amiridis[4], Nikos Mihalopoulos[1,2] and Evangelos Gerasopoulos[1]

[1] Institute for Environmental Research and Sustainable Development, National Observatory of Athens, Greece
[2] Environmental Chemical Processes Laboratory, Department of Chemistry, University of Crete, Greece
[3] Institute of Environmental Physics and Remote Sensing, University of Bremen, Germany
[4] Institute for Astronomy, Astrophysics, Space Applications and Remote Sensing, National Observatory of Athens, Greece
[5] Laboratory for Modeling and Observation of the Earth System (LAMOS), University of Bremen, Germany
[6] Climate and Atmosphere Research Center, CARE-C, The Cyprus Institute, Cyprus
[7] Physikalisch-Meteorologisches Observatorium Davos, World Radiation Center, Switzerland
[8] Physics Department, Kuwait University, Kuwait
[9] Laser Remote Sensing Laboratory, National Technical University of Athens, Greece

**Abstract.** In this study, we report on the retrieval of aerosol extinction profiles from ground-based scattered sunlight multi-axis differential optical absorption spectroscopy (MAX-DOAS) measurements, carried out at Athens, Greece. It is the first time that aerosol profiles are retrieved from MAX-DOAS measurements in Athens. The reported aerosol vertical distributions at 477 nm are derived from the oxygen dimer ($O_4$) differential slant column density observations at different elevation angles by applying the BOREAS retrieval algorithm. Four case studies have been selected for validation purposes; the retrieved aerosol profiles and the corresponding aerosol optical depths (AODs) from the MAX-DOAS are compared with lidar extinction profiles and with sun photometric measurements (AERONET observations), respectively. Despite the different approach of each method regarding the retrieval of the aerosol information, the comparison with the lidar measurements at 532 nm reveals a very good agreement in terms of vertical distribution, with r>0.85 in all cases. The AODs from the MAX-DOAS and the sun-photometer (the latter at 500 nm) show a satisfactory correlation (with r≈0.6 in three out of the four cases). The comparison indicates that the MAX-DOAS systematically underestimates the AOD in the cases of large particles (small Ångström exponent) and for measurements at small relative azimuthal angles between the viewing direction and the Sun. Better agreement is achieved in the morning, at large relative azimuthal angles. Overall, the aerosol profiles retrieved from MAX-DOAS measurements are of good quality; thus, new perspectives are opened up for assessing urban aerosol pollution on a long term-basis in Athens from continuous and uninterrupted MAX-DOAS measurements.



# 1 Introduction

Tropospheric aerosols originate from both natural and anthropogenic sources. The lifetime of aerosols in the troposphere ranges from a few days to a few weeks, depending on their size and meteorology (e.g. Pandis et al., 1995). They take part in
atmospheric processes through (i) nucleation and interaction with clouds (e.g. Twomey et al., 1977; Rosenfeld et al, 2014), (ii) participation in chemical and photochemical reactions, by providing the required surface for heterogeneous reactions to take place (Andreae & Crutzen, 1997) and (iii) absorption and scattering of incoming solar and earth's IR radiation, affecting atmospheric dynamics and stability (e.g. Dubovik et al., 2002) and the Earth's climate (IPCC, 2001). Significant decrease of UV-Vis irradiance reaching the ground due to urban aerosol pollution has been reported in various cases (e.g. Zerefos et al.,
2009; Chubarova et al., 2011).

According to a survey conducted in 25 large European cities, Athens occupies the third position on European level in exceedances of particle pollution regulations (Pascal et al., 2013). Saharan dust transported from the African continent is the main natural source of tropospheric aerosols in Athens (e.g. Kanakidou et al., 2007; Gerasopoulos et al., 2011; Raptis et al.,
2020), while common anthropogenic sources are traffic emission and domestic heating (Markakis et al, 2010; Gratsea et al., 2017). Wildfires also contribute to the aerosol mixture in the area occasionally, either by nearby events (Amiridis et al., 2012) or by long-range transport (Papayannis et al.; 2009, Amiridis et al., 2011; Mona et al., 2012). Whereas emissions of most air pollutants, such as $SO_2$, are expected to decrease by more than 80% until the end of the 21st century, the decrease of aerosol emissions is projected to be small (IPCC 2007) and thus aerosols may play an even more critical role in air quality in
the future. Therefore, long-term continuous measurements, providing information on the spatial and temporal distribution of aerosols, are of great importance to urban air pollution assessment and to the understanding of the aerosol contribution to Earth's climate. The knowledge of the vertical distribution of aerosols is necessary for understanding the mechanisms underlying the formation and development of urban smog.

Satellite, airborne and ground-based measurements are widely used to derive aerosol vertical profiles (e.g., Papayannis et al., 2005; Schmid et al., 2006; DeCarlo et al., 2008; Solanki and Singh, 2014); satellite measurements sometimes fail to be accurate in the lower atmosphere, while airborne measurements, although accurate in the lower atmosphere, are temporally restricted. In contrast, ground-based measurements can provide both a very good record of the lower troposphere and a satisfactory temporal resolution. However, since the ground-based profile measurements are mainly relying on lidar systems
(e.g., the European Aerosol Research Lidar Network - EARLINET – within the European Research Infrastructure for the observation of Aerosol, Clouds and Trace Gases - ACTRIS), they are costly in terms of setup and operation. An additional option for ground-based observations is the MAX-DOAS technique, which has been gaining ground over the last years (e.g., Wittrock et al., 2004; Heckel et al., 2005, Ma et al., 2013, Schreier et al., 2020) since it can provide low-cost, continuous and uninterrupted measurements without the need for absolute radiometric calibration. The MAX-DOAS technique has also been



shown to be very promising for the retrieval of aerosols' vertical distribution (e.g., Sinreich et al., 2005; Lee et al., 2009; Clémer et al., 2010; Wagner et al., 2011). In some studies, the retrieved aerosol extinction profiles were compared to the corresponding profiles derived from lidar (e.g., Irie et al., 2008; Zieger et al., 2011; Bösch et al., 2018) or Aerosol Robotic Network (AERONET) based measurements (e.g. Wang et al., 2016). For the Athens area, although several studies have been published on aerosol extinction profiles from lidar measurements (e.g. Papayannis et al., 1998; Matthias et al., 2004; Papayannis et al., 2005), vertical trace gas and aerosol profile retrievals from MAX-DOAS have not been published so far.

In the scope of this paper, a retrieval algorithm, recently developed by the Institute of Environmental Physics and Remote Sensing of University of Bremen (Bösch et al., 2018), is employed in order to obtain vertical distributions of aerosol extinction from $O_4$ MAX-DOAS measurements over the urban environment of Athens. $O_4$ is an atmospheric absorber with a known concentration profile, therefore its absorption measurements can be used as an indicator of the aerosol induced light path changes (Wagner et al., 2004).

For validation purposes the outcomes of our calculations are compared to established techniques; the retrieved profiles are compared to profiles from ground-based lidar measurements (EARLINET station) and the AOD to sun-photometer measurements (AERONET station).

A description of the instruments used in this study (location, instrumentation and data retrieval) along with a brief description of the profile retrieval algorithm are given in section 2. In section 3, we present the derived aerosol vertical distributions for four selected case studies and we compare the MAX-DOAS aerosol extinction coefficient profiles and the AOD with lidar and sun-photometric measurements, respectively. The findings are summarised in section 4, where also the conclusions of this study are provided.

## 2 Methodology

### 2.1 Location

Four mountains surround the city of Athens, forming a basin that is open to the south and southwest. This special topography plays an essential role in the accumulation of atmospheric pollutants over the city under certain meteorological conditions (Kassomenos et al., 1995). Moreover, dust transport episodes from North Africa also contribute to the aerosol load of the city (e.g. Gerasopoulos et al., 2009; Kosmopoulos et al., 2017). In general, the Athens area can be considered as an example of various aerosol types such as dust, local pollution, marine, biomass combustion and their mixtures (Soupiona et al., 2019).

Figure 1 shows the Greater Athens area and location of each instrument used in this study. The MAX-DOAS instrument is located at the premises of the National Observatory of Athens (NOA, 38.05° N, 23.86° E, 527m a.s.l), to the north of the city. No strong emission sources are present around the measurement area, which is considered as suburban background. The



lidar system performs measurements at the National and Technical University of Athens (NTUA, 37.97 ºN, 23.79 ºE, 212m a.s.l.) and the site is considered as suburban background. The CIMEL sun-photometer is installed at the premises of NOA at Thissio hill (37° 58′ N, 23° 43′ E, 150m a.s.l.), which, despite being located in the city centre, is considered as urban background (Paraskevopoulou et al., 2015). Information about the instruments is provided in Table 1.

## 2.2 Instrumentation and data retrieval

### 2.2.1 MAX-DOAS

The MAX-DOAS instrument employed in this study is part of the BREDOM network (Bremian DOAS network for atmospheric measurements, http://www.iup.uni-bremen.de/doas/groundbased__data.htm) and has been operating continuously since October 2012. It comprises a grating spectrometer (LOT 260S, 600 l/mm ruled grating) connected via an optical fiber bundle to a computer-controlled telescope unit. The spectrometer covers a spectral range from 330 to 500 nm with a spectral resolution of approximately 0.7 nm. The detector used is a CCD (Charge-Coupled Device) by Andor Technology, with 2048 x 512 pixel resolution, cooled to -40ºC.

The telescope performs intensity measurements at eight elevation angles (-1º, 0º, 1º, 2º, 4º, 8º, 15º, 30º), as well as to the zenith. Measurements in eight azimuthal directions are performed, but in this study, only the S direction, pointing at 52.5º (with respect to South) and representing the urban area of the city (Gratsea et al., 2016), is considered (Fig. 1). The duration of one full scanning cycle (azimuthal and elevation scanning) is about 15 min, thus about 30 measurement cycles per day are available in winter and 45 in summer.

The spectral measurements are analysed using the DOAS technique; the Beer-Lambert law is considered as the solution of the radiative transfer equation (Hönninger and Platt, 2002; Wittrock et al., 2004) and the absorption spectrum is separated into broad and narrow spectral features that show low and high frequency variations, respectively, as a function of wavelength. The narrow spectral features correspond to the unique narrow-band absorption structures of the trace gases, while the broad ones represent the attenuation of solar radiation by scattering processes in the atmosphere as well as the continuum absorption by trace gases and the instrument. For the derivation of the slant column density (SCD, defined as the concentration of the absorber integrated along the light path), a polynomial accounting for the broad spectral features and the laboratory cross-sections of the retrieved species are fitted to the measured optical depth. To determine the optical depth, the logarithm of the ratio of the current horizon measurement (I) and the reference intensity ($I_o$) is taken.

The SCD of the oxygen dimer ($O_4$), i.e. the slant optical thickness of the absorber divided by the absorption cross section, measured at different elevations is used as input to the retrieval algorithm for the calculation of the aerosol distribution. The slant column of the $O_4$, a weak molecular absorber with a well-known vertical profile (the $O_4$ concentration is proportional to



the square of the $O_2$), is almost linearly dependent on the average photon pathlengths (Pfeilsticker et al., 1997) and thus can
be used as an indicator of the presence of clouds or aerosols in the atmosphere. The $SCD_{O4}$ is calculated by fitting to the
measured optical depth the laboratory spectrum of $O_4$ (Hermans et al., 2003), $NO_2$ (Vandaele et al., 1998) and of $O_3$
(Bogumil et al., 2000) and a polynomial of degree 4 which accounts for the broad spectral features. The fitting spectral
window used is 425-490 nm. In order to retrieve the tropospheric $SC_{O4}$, the zenith observation, corresponding to each
measurement cycle, is used as the reference measurement $I_o$, canceling in this way the Fraunhofer lines in the solar spectrum
and the stratospheric contributions to the SCD.

The differential tropospheric vertical column densities (VCD) of $NO_2$, shown in section 3.1, derived by using air mass
factors (AMF) calculated with the SCIATRAN radiative transfer model (Rozanov et al., 2000). To convert the differential
tropospheric SCD to the corresponding tropospheric VCD, the differential AMF ($AMF_\alpha - AMF_{90^o}$) is required, namely the
difference between the AMF at the same elevation α as the SCD measurement and the AMF at the zenith (Eq. 1).

$$VCD = \frac{SCD_\alpha}{(AMF_\alpha - AMF_{90^o})} \qquad (1)$$

The AMF describes the weighting of the absorption as a function of the relative azimuth and the solar zenith angle (SZA) for
a given atmospheric profile and at a specific wavelength.

### 2.2.2 EOLE lidar system

The six-wavelength Raman-backscatter lidar system (EOLE) operates in Athens since February 2000 as part of the
EARLINET network (Pappalardo et al., 2014). The system is designed following the optical set-up of a typical member
station (Kokkalis 2017), meeting all the quality assurance requirements of the network. The emission unit is based on a
pulsed Nd:YAG laser, emitting high energy pulses at 355, 532 and 1064 nm with a repetition rate of 10 Hz. The optical
receiver is based on a Cassegrainian telescope (600 mm focal length and a clear aperture diameter of 300 mm), directly
coupled with an optical fiber, to the wavelength separation unit, detecting signals at 355, 387 ($N_2$ Raman line of 355nm), 407
($H_2O$ Raman line of 355nm), 532, 607 ($N_2$ Raman line of the 532nm) and 1064 nm. For every measuring cycle 1000 lidar
signal returns are stored (every ⁓1.66). For each case presented in this study, we used hourly averaged profiles, which
correspond to approximately 34 individual signal acquisitions (Kokkalis et al., 2012).

During day time operation, the system is capable of providing aerosol backscatter profiles ($\beta_{aer}$) at 355, 532 and 1064 nm,
based on the standard backscatter lidar technique and employing the Klett inversion method (Klett, 1981). This technique
assumes the existence of an aerosol-free region (e.g. upper troposphere) and requires an a-priori assumption of the lidar ratio
value (the ratio of the extinction to backscatter coefficient, $S_{aer}$). A variety of studies revealed a wide range for the lidar
ratios, covering values from 20 to 100 sr (Ackermann, 1998; Mattis et al., 2004; Amiridis et al., 2005; Müller et al., 2007;
Papayannis et al., 2008; Groß et al., 2011; Giannakaki et al., 2015). When the elastic backscatter lidar technique is used, the





assumption of a constant lidar ratio value throughout the laser sounding range, becomes very critical when solving the lidar equation; in this case, the overall uncertainty, including both statistical and systematic errors, on the retrieved $\beta_{aer}$ values, is of the order of 20–30% (e.g. Rocadenbosch et al., 2010). In this study, in order to account for the lidar ratio error assumption, we considered a lidar ratio input value of $50 \pm 20$ sr. All the lidar profiles where obtained with the Single Calculus Chain (SCC) processing platform (D'Amico et al., 2016; Mattis et al., 2016), which is developed in the framework of EARLINET to ensure the high-quality products of the network, by implementing quality checks on both raw lidar data and final optical products.

One of the lidar's main limitations is the distance of full overlap between the laser beam and the receiver's field of view, which makes difficult for the instrument to obtain useful and accurate aerosol-related information below that height. Wandinger and Ansmann (2002) demonstrated that when not applying overlap correction in lidar signals, the retrieved aerosol extinction coefficient may take even non-physical negative values for heights up to the full overlap. The incomplete overlap effect can be solved by using Raman measurements under night-time conditions. In this study, only daytime measurements are used and therefore no overlap correction is applied on the signals. The geometrical configuration of EOLE results in full overlap distance of 500-800 m above ground (Kokkalis 2017). The aerosol extinction values below the 1000 m a.s.l. height are considered inside the overlap region and therefore were omitted from the extinction profile comparison. Nevertheless, to calculate the AOD from the lidar profiles, the lowermost trustworthy value of the extinction coefficient was assumed constant (height-independent), and also representative of the aerosol load in the overlap region.

### 2.2.3 CIMEL sun-photometer

The reported columnar aerosol optical properties have been retrieved by a CIMEL sun-photometer (Holben et al., 1998). The instrument is part of NASA's global sun photometric network, AERONET, and performs automatic measurements of the direct solar radiance at the common wavelengths of 340, 380, 440, 500, 675, 870, 940 and 1020 nm every 15 min and diffuse sky radiance at 440, 675, 870 and 1020 nm. These measurements are further used to provide both optical and microphysical aerosol properties in the atmospheric column (Dubovik et al., 2006). The CIMEL data used in this study are the cloud screened and quality assured level 2.0 data products, providing information about the columnar AOD and the Ångström exponent. The AOD uncertainty is $<\pm0.02$ for UV wavelengths and $<\pm0.01$ for wavelengths larger than 440 nm (Eck et al., 1999).

### 2.3 BOREAS profile retrieval algorithm

The BRemen Optimal estimation REtrieval for Aerosol and trace gaseS (BOREAS) is an optimal estimation based profile retrieval algorithm developed at the Institute of Environmental Physics, University of Bremen (Bösch et al., 2018). Slant column densities of $NO_2$ and $O_4$ from MAX-DOAS measurements at different line of sight (LOS) directions, as well as climatology profile files are used as inputs. The BOREAS algorithm is based on the SCIATRAN radiative transfer model


(Rozanov et al., 2005), which is used to calculate box-air-mass-factors and weighting functions, needed for the profile inversion. For the radiative transfer model (RTM) calculations, scattered light in a spherical atmosphere (multiple scattering) and atmospheric profiles of pressure and temperature from the U.S. Standard Atmosphere (NASA, 1976) are considered. The aerosol inversion problem is expressed through the minimisation of Eq. (2):

$$\left\| \Delta\tau(\lambda,\boldsymbol{\Omega}) - \Delta\tilde{\tau}(\lambda,\boldsymbol{\Omega},N_\alpha(z)) - P(\lambda,\boldsymbol{\Omega}) \right\|^2 \rightarrow min \tag{2}$$

,where $\Delta\tau$ denotes the measured $O_4$ differential slant optical thickness, $\Delta\tilde{\tau}$ the simulated differential slant optical thickness, $\boldsymbol{\Omega}$ the measurement geometry, $N_\alpha(z)$ the a priori aerosol number concentration profile and P a polynomial of lower order. Since the relationship between the concentration profile and the $O_4$ differential slant optical depth is not linear, an iterative Tikhonov regularisation technique, along with weighting function matrices expressing the sensitivity of the measurements to changes in the aerosol profile, are used for the solution of the optimisation problem.

The temporal resolution of the measurements is about 15 minutes, which corresponds to the duration of one full scanning cycle through all directions over the city. The vertical sampling of the retrieved profile is 0.05 km, with the bottom layer considered at the sea level and the top layer at 4 km a.s.l. The AOD is calculated by integrating the BOREAS retrieved aerosol extinction coefficient vertically. Table 2 summarises the parameter settings used for the BOREAS retrieval. More details about the values assigned to each parameter are given in section 3.2.

## 3 Results and discussion

### 3.1 Selected case studies

The main objective of this study is to assess the retrieved aerosol profiles from MAX-DOAS measurements by comparing them with well established sun-photometric measurements (CIMEL) and lidar retrievals. Therefore, certain cases had to be selected with available and valid data from all three instruments. Additionally, the selected cases had to coincide with cloud-free days, as all of the used measurement techniques have more subtantial uncertainties in the presence of clouds. During the period from January 2015 to June 2016, four cases were found to meet the above conditions, covering winter, summer and spring: i) 05 February 2015 under the influence of a weak dust event, ii) 09 July 2015 with enhanced morning levels of $NO_2$ for this season (Gratsea et al., 2016) iii) 10 July 2015 with typical levels of pollution and iv) 04 April 2016 with enhanced levels of $NO_2$. In order to identify the sources of air masses reaching Athens on the specific dates, 4-day air mass back trajectories at different altitudes, calculated using the NOAA-HYSPLIT (Hybrid Single-Particle Lagrangian-Integrated Trajectory) model (Draxler and Hess, 1997) were used. Potential for Saharan Dust transport below 4 km, which is the highest point of our retrievals, was identified only for case (i) (Fig.2). In the rest of the cases, the air masses below 4 km originate from N/NE directions, and are thus not associated with dust aerosols. The $NO_2$ levels, measured by MAX-DOAS and presented in Fig. 3, are used as an indicator for the pollution levels over the city. The mean diurnal $NO_2$ DSCDs for



winter and summer months, as reported by Gratsea et al. (2016), range from $6 \cdot 10^{16}$ to $9 \cdot 10^{16}$ and $5 \cdot 10^{16}$ to $11 \cdot 10^{16}$
molec/cm$^2$, respectively. Thus, enhanced pollution levels are observed during the morning hours in cases (ii) and (iv). The absence of clouds is established using in-situ meteorological observations from the monitoring station of the National Observatory of Athens at the centre of the city and is also verified by the MAX-DOAS retrieved $O_4$ slant columns throughout the day. The above mentioned cases will henceforth be referred to as cases (i), (ii), (iii) and (iv), respectively.

The MAX-DOAS elevation angle sequence includes not only measurements above the horizon, but also at 0° and -1° elevation. Initially, the aerosol extinction profiles from MAX-DOAS were derived using measurements including these elevation angles. However, experience showed that better results are achieved only when elevation angles above the horizon are used for the inversion, most probably due to the negative impact of an inaccurate spectral surface reflectance on the retrieval. Thus the -1° and 0° elevation angles were excluded from the calculations. With this choice, little information is
available for the profile retrieval below the station altitude of 527 m a.s.l. As already mentioned (Fig. 1), the S azimuthal direction is associated to the city's urban atmospheric conditions (Gratsea et al., 2016). The S direction also coincides with the sun-photometer's location and is also close to the lidar's measurement site (Fig.1).

### 3.2 Aerosol extinction vertical profile retrievals

MAX-DOAS measurements and the BOREAS retrieval algorithm were used for the calculation of the diurnal aerosol
extinction vertical distribution over the urban (S) area (Fig. 4) for the selected case studies. Single scattering albedo (SSA) and phase functions are not retrieved in BOREAS and have to be prescribed. Therefore AERONET measurements are used for specifying SSA ($\omega$) and asymmetry factor (g) values. However, $\omega$ and g were not available in AERONET data for case (iv), therefore in this case the algorithm was run with the monthly mean of SSA ($\omega$=0.91) and asymmetry factor (g=0.68) from the following year, as derived from the AERONET data (Table 2). A fixed surface albedo ($\alpha$=0.15), based on a
previous study for Athens (Psiloglou et al., 2009), was used in all cases.

The results for case (i) reveal a significant variation of the aerosol distribution in the vertical direction. The maximum retrieved extinction values in this case, reach almost 0.3 km$^{-1}$ close to the surface until noon and after that time the aerosol load declines. This temporal variation can be attributed to changes in the prevailing wind speed and direction throughout the
day; as recorded by NOA's meteorological monitoring station at Thissio, the prevailing wind direction from 09:00LT until 12:00LT (LT=UTC+2 winter time and UTC+3 summer time) was from the south with speed from 1 to 4 m/s, while easterly winds with speed reaching 10 m·s$^{-1}$ started blowing at 13:00 LT, efficiently ventilating the Athens basin and removing the dust and atmospheric pollutants. As shown in previous works conducted in the area (e.g Fourtziou et al., 2017), wind speed below 3 m·s$^{-1}$ favours the accumulation of pollutants.






The two cases, (ii) and (iii), present two aerosol layers; a surface layer and an elevated one extending up to 3 km between 13:00 and 17:00LT. Lidar retrievals also show an elevated extinction layer in both cases, as discussed in section 3.3. However, the separation of the two layers could be an artifact which arises from the fact that the MAX-DOAS retrieval's response to a box-like distribution (e.g. a well developed planetary boundary layer - PBL) leads to slight oscillations around

this box due to the a priori smoothing. Both cases are related to weak prevailing winds (<4 m·s⁻¹), which favour the development of a vertically extended aerosol layer. In case (iii) the near-surface aerosol load starts building up earlier than in case (ii). The higher aerosol load in case (iii) is also corroborated by sun photometric measurements, which are presented and discussed in section 3.4.

A homogeneous surface aerosol layer with low levels of aerosol extinction (0.2 km⁻¹ maximum value) is present over the urban area throughout the whole day in case (iv). The height of the aerosol layer is around 800 m.a.s.l. Given that the $NO_2$ level, characteristic of anthropogenic pollution, is high during this day (Fig. 3), higher particle pollution levels would be expected.

### 3.3 MAX-DOAS aerosol extinction profiles evaluation

The BOREAS retrieved aerosol extinction profiles from the MAX-DOAS measurements at 477 nm, between 0.5 km (station's elevation) and 4 km height, are compared with the lidar aerosol extinction coefficient measurements at 532 nm, between 1 and 4 km height, for the selected case studies (Fig. 6). Representative morning and afternoon snapshots during each day have been chosen to be presented and discussed. The lack of morning profiles for some days is due to the absence of lidar data, thus, both morning and evening data is available only for cases (i) and (ii). The lidar profile used in each figure

is the average of all profiles retrieved between the starting and the ending time of the presented MAX-DOAS profiles. The uncertainty in the lidar extinction profiles increases substantially for altitudes below 1000 m.a.s.l. due to the loss of overlap between the telescope field of view and the laser beam (Wandinger and Ansmann, 2002; Kim et al., 2008, Papayiannis et al., 2008); hence the lidar data for altitudes below 1000 m a.s.l. is not presented and only measurements above 1000 m a.s.l. are considered for the calculation of the correlation between the two instruments. Another point that has to be considered when

comparing the results from the two instruments is that the lidar profiles are characterised by high vertical and temporal resolution and degradation to the sensitivity of the MAX-DOAS profiles is necessary in order to have a meaningful comparison to the MAX-DOAS data. According to the method described by Rodgers and Connor (2003), the degraded lidar profile $x_f$ can be estimated by applying the equation

$$x_f = x_a + AK \cdot (x - x_a) \tag{3}$$

with $x_a$ being the a priori profile used in the algorithm calculations, x the initial lidar profile and AK the averaging kernel from the BOREAS retrieval. The averaging kernel (Fig. 5) denotes the sensitivity of the retrieved profile to the true atmospheric profile for each layer and in fact it represents the smoothing of the true profile in the retrieval. The lidar profile, degraded to 50 m vertical resolution, represents the MAX-DOAS profile that would have been retrieved, if the true



extinction profile was x. Last, but not least, the horizontal distance (13 km) between the two measurement sites and the different operation principles of the two instruments should be noted. The lidar system retrieves information from the air mass right above the measurement site, while MAX-DOAS probes air masses along the line of sight of the telescope pointing from the top of a hill towards the city centre. Thus, some discrepancies are expected, especially when the aerosol pollution is not horizontally homogeneous over the Athens basin, and the comparison is mainly focused on a qualitative basis.


Each case is examined separately and some performance statistics - correlation coefficient (r), median lidar/MAX-DOAS ratio, root mean square error (RMSE), fractional gross error (FGE) - are shown in Table 3. This set of statistics has been chosen as suitable to provide a detailed view of the algorithm performance; it has been proposed (Morris et al., 2005) that a FGE less than or equal to 0.75 is a criterion to evaluate good performance of an algorithm, therefore, any FGE>0.75 is used

as indicator of a relatively poor performance in this study. In order to perform the statistics calculations we averaged the four MAX-DOAS profiles comprising each case. Thus, all performance statistics have been calculated using the temporally averaged MAX-DOAS profile for each case and the corresponding degraded lidar profile, so that both profiles are of the same temporal and vertical resolution. In all cases, 61 data points are used for the derivation of the statistics.

Case study (i)
       In case (i), the two instruments seem to be in excellent agreement, in terms of both extinction levels - up to 3 km - and correlation, when the morning measurements are considered; the median ratio is close to unity and the correlation coefficient is very high (r=0.98). In the afternoon, a peak ( ~0.15 km$^{-1}$) at 1.5 km is recorded by the MAX-DOAS in contrast to the lidar profile, which shows no elevated aerosol layer, leading to lower - still high - correlation (r=0.84). However, the large

discrepancy between the original and the degraded lidar profile is attributed to the fact that the AKs of the afternoon retrievals illustrate low sensitivity of the retrieved profile to the true atmospheric profile for altitudes up to 2.5 km (Fig. 5).
       It should be mentioned that this is the only case in the present study, where high aerosol load is found in the upper levels (free troposphere) in the original lidar profiles due to transboundary transport of aerosols at higher altitudes. The fact that, at these altitudes, the MAX-DOAS only agrees with the degraded lidar profiles (which means after including the AK

information) suggests more significant errors in the a priori aerosol profiles and the reduced capacity of the MAX-DOAS to capture the characteristic inhomogeneity at higher atmospheric layers during aerosol transport episodes. Nevertheless, an overall satisfactory performance of the algorithm is indicated by the FGE (0.6).

       Case study (ii)
The retrieved MAX-DOAS profiles agree quite well with the degraded lidar profiles; they both show an aerosol layer extending up to about 2.5 km and the correlation coefficient is very high (r ≈ 0.9), during both morning and afternoon measurements. In the afternoon, however, the MAX-DOAS measurements result in higher extinction levels by almost 75%



compared to the degraded lidar profile. As shown in Fig. 6 (middle row panels), in this case MAX-DOAS tends to overestimate the lidar extinction levels mainly at higher altitudes, a fact that can be attributed to the smoothing effect of the

retrieval procedure on the true profile; given that a MAX-DOAS profile algorithm cannot retrieve sharp edges, the underlying narrow high altitude enhancement in the afternoon propagates through the retrieval into a smoother and broader aerosol peak. The FGE, ranging from 0.45 to 0.55, indicates a good performance of the algorithm.

Case study (iii)

The two instruments seem to correlate very well (r=0.97). The MAX-DOAS coincides well with the aerosol extinction levels from the lidar measurements, especially in the first 2 km; the lidar to MAX-DOAS ratio is equal to 0.91. Nevertheless, when the original lidar profile is considered, a clear discrepancy in the extinction levels is present; the lidar peak value (0.16 km$^{-1}$) is enhanced by a factor of two. The discrepancy between the original and the degraded lidar profile results from the low sensitivity of the averaging kernels for heights up to about 2 km (Fig. 5, case (iii)), which plays significant role in the

degradation (smoothing) of the lidar retrieval. The RMSE is small (0.01 km$^{-1}$) and the low FGE (0.23) indicates good performance of the algorithm.

Case study (iv)

The profiles resulting from both instruments display an aerosol layer of about 1.5 km deep. The profile shapes are very

similar and the correlation coefficient (r=0.86) indicates a high correlation between the two instruments. However, the MAX-DOAS underestimates by almost 50% the lidar aerosol extinction (median lidar/MAX-DOAS ratio=1.8). Although the correlation is high and the RMSE is small (0.02), the relatively high FGE (0.77) indicates a poor performance of the algorithm for the specific case. This FGE value, however, results from the high median ratio of the two profiles, which in turn results from the low extinction levels, since the absolute difference between the two profiles is not that large.


Overall, the correlation between lidar and MAX-DOAS measurements is very good (0.84 < r < 0.98) in all cases and a good agreement in the profile shape and altitude of the peak extinction level is also observed. The failure of the MAX-DOAS to capture clearly distinguished aerosol layers is attributed to the smoothing effect due to the presence of a priori constraints during the retrieval procedure. The RMSE ranges from 0.01 to 0.05 km$^{-1}$ in all cases. A satisfactory fractional gross error

(<0.60), indicating a very good performance of the MAX-DOAS profiling algorithm, has been calculated in all cases with average or high extinction values. A higher FGE (0.77) has been calculated only in the case of very small extinction levels. Due to the different operation principles of each instrument (active/passive remote sensing), the different wavelengths and the different air masses probed by each instrument, a full agreement in the derived profiles would not be expected. In particular, the lidar profiles represent the aerosols which are directly over the measurement site, whereas the MAX-DOAS

profiles are representative of the atmosphere at a distance of several kilometres along the line of sight of the instrument. Another conclusion arising from these four cases is that the MAX-DOAS fails to detect part of the urban aerosol pollution



when the pollution levels are low (e.g. case(iv)) and also fails to capture the inhomogeneity at higher altitudes in case of aerosol transport episodes.

**3.4 AOD evaluation**

In this section, we report on the comparison between the retrieved aerosol optical depth (AOD) from MAX-DOAS measurements at 477 nm and from CIMEL measurements at two wavelengths (440 nm and 500 nm), during the aforementioned case studies (Fig. 7). When looking at the figures, one should consider that the CIMEL AOD uncertainty is estimated to be approximately 0.01 for wavelengths > 400 nm (Eck et al., 1999). The MAX-DOAS AOD uncertainties are shown in the figures. The Ångström exponent, derived from the CIMEL measurements (400 - 870 nm), is also taken into

account, as a qualitative indicator of aerosol particle size, in order to investigate the origin of the aerosols (natural-dust or anthropogenic sources) and the performance of the MAX-DOAS retrievals for different aerosol types and sizes. An overview of the comparison statistics (described in section 3.3), representative of the degree of agreement between MAX-DOAS and CIMEL measurements at 500 nm, is presented in Table 4. The calculations were made on hourly basis to achieve uniform results regarding the air masses.


Case study (i)

The very small Ångström exponent, ranging between 0.05 and 0.13 throughout the day, indicates the dominance of coarse particles in the aerosol distribution. Given the cloud-free sky conditions and the potential for dust transport found for this day by the NOAA-HYSPLIT (Fig. 2), these particles are probably associated with the presence of dust in the atmosphere.

Although the correlation between the two instruments is moderate (r=0.56), the calculated AOD levels from the MAX-DOAS measurements agree very well with the CIMEL measurements (median ratio CIMEL/MAX-DOAS = 1.08). The daily averaged AOD values are 0.38 (± 0.02)  and 0.39 (± 0.03)  for MAX-DOAS and CIMEL, respectively, and much higher than the climatological monthly average value (0.27 ± 0.03) for February in Athens, as reported in Gerasopoulos et al. (2011). The RMSE is small (0.03) and the FGE, which is close to zero (0.09) implies excellent performance of the algorithm. The

moderate correlation results may arise from the fact that CIMEL performs direct sun measurements, whereas MAX-DOAS measurements - and the subsequent AOD retrieval - are performed at a fixed azimuthal direction. Thus, the CIMEL measurements are highly affected by variations in the temporal and spatial distribution of the aerosols. The AOD from the lidar measurements at 12:00 and 15:00LT (0.37 and 0.35, respectively) coincides well both with CIMEL and MAX-DOAS measurements.


Case study (ii)

In this case, the large values of the Ångström exponent ($\alpha \geq 2$) are indicative of the presence of fine mode aerosols that are associated with urban pollution (Westphal and Toon, 1991, Eck et al., 1999, Gerasopoulos et al., 2011). The considerable levels of $NO_2$ measured during this day (Fig. 3), indicate the presence of anthropogenic pollution. The two instruments are



again moderately correlated (r=0.63), however if the afternoon measurements - after 16:00LT - are excluded, the correlation becomes very good (r=0.86). The median ratio (0.72) indicates that the MAX-DOAS overestimates the AOD levels measured by the CIMEL. However, the overestimation is more profound during the morning, while in the afternoon the MAX-DOAS slightly underestimates the measured AOD by about 10%, a fact that can be attributed to inaccuracies based on the geometry of the light scattering in the forward direction. The daily averaged AOD values are 0.23 and 0.19 for MAX-

DOAS and CIMEL, respectively. The small values of both the RMSE (0.09) and the FGE (0.35) are indicators of very good performance of the algorithm. The gaps in the CIMEL data in this case, as well as in case (iii), are probably due to saturation of the instrument. The lidar AOD in this case (0.21 at 13:00LT and 0.26 at 16:00LT) agrees well with the CIMEL measurements at 500 nm, but is lower than the AOD from the MAX-DOAS; the difference is more considerable in the afternoon.


#### Case study (iii)

In the third case study, the measurements from the two instruments seem to be in better agreement during morning hours. Overall, the MAX-DOAS overestimates the AOD levels from CIMEL by 20%, however the underestimation due to light scattering geometry is about 25% after 13:00UTC (16:00LT). The moderate results with respect to correlation (r=0.55) are

due to the large discrepancy between the two instruments during the afternoon. If only the measurements until 16:00LT are considered, the correlation is considerably improved (r=0.75). Despite the non-satisfactory correlation, the calculated FGE (0.32) indicates a very good performance of the algorithm. The lidar-derived AOD in the afternoon coincides very well with the MAX-DOAS measurement. Unfortunately, no CIMEL or lidar data are available around noon in order to validate the aerosol plume captured by MAX-DOAS. It should be noted, though, that case studies (ii) and (iii) (both summer days in

July) exhibit the same diurnal pattern; lower values in the morning, steadily increasing throughout the day and then slightly declining in the afternoon. A similar diurnal AOD pattern was found for summer in Athens by Gerasopoulos et al., 2011 and this pattern has been associated with local urban or industrial sources (Smirnov et al., 2002).

#### Case study (iv)

The Ångström exponent in this case (α≈1) indicates the presence of coarse aerosols (radii≥0.5 μm) in the atmosphere (Westphal and Toon, 1991, Eck et al., 1999). The NOAA HYSPLIT backtrajectories show the potential for African dust transport to Athens, however, at higher altitudes of up to 5 km. Nevertheless, despite the presence of coarse particles, the AOD levels are low; the daily averaged AOD values are 0.15 and 0.19 for MAX-DOAS and CIMEL, respectively. The MAX-DOAS underestimates the AOD with respect to CIMEL by about 15% (and by 30% if only the afternoon

measurements are considered), the RMSE is 0.05 and the FGE is 0.27, yet it seems that the two measurement techniques are not correlated (r=-0.44). However, the comparison in terms of correlation, results in better outputs if only the morning measurements are considered; in this case the correlation coefficient is 0.80. It seems that MAX-DOAS, in this case, fails to



detect the accumulation of coarse particles. Nevertheless, the lidar measurements (AOD=0.15) in the afternoon agree very well with the MAX-DOAS AOD (0.14).


Overall, a systematic underestimation of the AOD, by 10 to 30%, by the MAX-DOAS is observed in the afternoon measurements, when the relative azimuthal angle between the MAX-DOAS viewing direction and the Sun is small. Better agreement is achieved at large relative azimuthal angles in the morning. This finding has also been reported by Frieß et al. (2016) when comparing different retrieval algorithms with sun photometer measurements. The sun-photometer is located

downtown (150 m a.sl.) at lower altitude than the MAX-DOAS (527 m a.s.l.), thus it is more sensitive to aerosols in the lower troposphere. Nevertheless, the MAX-DOAS seems to detect well the typical urban aerosols in the boundary layer; the mean AOD difference (CIMEL minus MAX-DOAS) of all the measurements is 0.05 with standard deviation 0.09. Furthermore, CIMEL is a direct sun photometer, which means that in each measurement different air masses are detected, while the MAX-DOAS always points at the same direction; this operational difference is reflected in the non-satisfactory

correlation. When fixed values of SSA and asymmetry factor (instead of AERONET data) are used by BOREAS, it seems that MAX-DOAS fails to detect accumulated coarse particles (e.g. case (iv)), leading to underestimation in case of small Ångström exponent values (< 1). Friess et al. (2017) have also come to this conclusion during the CINDI-2 campaign. However, the underestimation could also be attributed to the high altitude aerosol layer detected by the lidar (Fig. 6); the MAX-DOAS' sensitivity at higher altitudes is low, thus it records lower AOD levels. It should also be noted that the

standard AERONET version 2 algorithm uses an $NO_2$ climatology with a spatial resolution much higher than the Athens city area (Giles et al., 2019), hence in certain cases the difference between the MAX-DOAS and the higher CIMEL AOD levels at 440 nm could result from additional $NO_2$ content in the atmosphere, which is the case of highly polluted days. The lidar-derived AOD levels coincide with the MAX-DOAS measurements in all cases, apart from the case (ii), in which MAX-DOAS overestimates both CIMEL and lidar measurements.

**4. Summary and conclusions**

An assessment of the retrieval of aerosol extinction profiles and AOD from MAX-DOAS measurements is presented for the first time for the urban environment of Athens. The profiling results are compared to lidar extinction profiles and to AODs obtained from sun photometric measurements. The intercomparison results are very promising, showing that the MAX-DOAS measurements provide a good estimation of the aerosol vertical profile over Athens. Although this intercomparison is

of great importance for the validation of the MAX-DOAS retrieval, the different operation, characteristics and measurement principles of each instrument, in addition to some comparison restrictions, have to be considered.

Regarding the spatial characteristics, (i) the measurements with the MAX-DOAS technique represent an area that includes the AERONET and the lidar locations, but it is not limited to them. Regarding the vertical aerosol information, (ii) the



MAX-DOAS retrievals are representative for 500-4000 m a.s.l., while the lidar profiles are valid for altitudes higher than 1000 m above the station and finally (iii) the AOD AERONET measurements describe the columnar aerosol properties representative of an area ranging from few up to 10 km radius above the Athens area, depending on solar elevation. Also, (v) the sun-photometer AOD observations probe the extinction in the full atmospheric column while MAX-DOAS retrievals are sensitive only to the lowest kilometers, leading to differences in the presence of aerosol layers at altitudes above 4 km.

Nevertheless, despite the comparison restrictions and the differences of the three instruments, the comparison of the retrieved profiles and the AODs shows that the MAX-DOAS measurements bode well for the future of aerosol measurements and they are able to provide a good estimation of the aerosol vertical distribution over Athens.

The vertical profiles retrieved by BOREAS profiling algorithm applied to the MAX-DOAS measurements are qualitatively

in good agreement with the lidar profiles smoothed with the MAX-DOAS averaging kernels; there is good agreement in aerosol layer shape and aerosol extinction levels in most of the cases. Very good correlation (r > 0.84) was found in all cases. A satisfactory fractional gross error (0.25 < FGE < 0.60) has been calculated in all cases with average or high extinction levels, indicating a good performance of the BOREAS profiling algorithm in these cases. In some cases, the observed underestimation of the aerosol extinction (by 10 to 30%) by the MAX-DOAS at small relative azimuth angles can be

attributed to the geometry of Mie scattering in relation to the location and viewing geometry of MAX-DOAS, resulting in MAX-DOAS' failure to detect part of the urban aerosol pollution. Overall, the agreement between the two instruments is encouraging, especially when considering the different nature of each technique and the different instrument locations, suggesting that the MAX-DOAS can accurately enough represent the aerosol vertical distribution.

The MAX-DOAS retrieved AODs show satisfactory agreement with the sun photometric measurements, in terms of AOD levels. The MAX-DOAS underestimates the AOD in the presence of coarse particles; CIMEL/MAX-DOAS ratio > 1 coincide with Ångström exponent values < 1. A systematic underestimation by MAX-DOAS is observed in the afternoon measurements due to MAX-DOAS' viewing geometry. Overall, the MAX-DOAS can be considered as an effective mean for measuring the aerosol levels in Athens; the average AOD difference of all measurements between the two instruments is

0.05. It is important to note that in Athens, a highly populated and polluted area, horizontal gradients, especially in anthropogenic aerosols, are very likely to occur, resulting in different air masses detected by each instrument and subsequently in discrepancies between MAX-DOAS and CIMEL measurements.

This intercomparison is of great importance for the validation of the MAX-DOAS retrieval, despite the different operation,

characteristics and measurement principles of each instrument. While evaluating the MAX-DOAS performance in tracking aerosols (AOD and vertical distribution) the following restrictions should be kept in mind: (i) the measurements with the MAX-DOAS technique represent an area that includes the AERONET and the lidar locations, but it is not limited to them. Regarding the vertical aerosol information, (ii) the MAX-DOAS retrievals are representative for 500-4000 m a.s.l., while the

lidar profiles are valid for altitudes higher than 1000 m above the station and finally (iii) the AOD AERONET measurements describe the columnar aerosol properties representative of an area ranging from few up to 10 km radius above the Athens area, depending on solar elevation. Also, (v) the sun-photometer AOD observations probe the extinction in the full atmospheric column while MAX-DOAS retrievals are sensitive only to the lowest kilometers, leading to differences in the presence of aerosol layers at altitudes above 4 km.

Despite the mentioned limitations, this work demonstrates that the MAX-DOAS measurements in Athens and the BOREAS algorithm can provide a good estimation of the aerosol vertical structure of the urban atmosphere, on a continuous and long-term basis, offering a reliable data set for scientific studies. There is certainly more work to be conducted in future studies in order to understand the sensitivity of the MAX-DOAS aerosol measurements based on different aspects of urban pollution evolution and long range transported aerosols.

**Data availability**

All data sets used and produced for the purposes of this work are freely available and can be requested from the corresponding author.

**Author contribution**

MG, PK and AR conceived the presented idea. MG performed the analysis and prepared the manuscript. TB developed the profile retrieval algorithm and provided guidance to MG on the algorithm calculations and parameterization. AR provided guidance to MG on MAX-DOAS data retrieval and profile calculations. MG designed the figures with support from PK and TB. AR and SK advised MG on the results interpretation. PK, AP and MM provided the lidar data. VA provided the sun-photometer measurements. AT made comments on the sun-photometer/max-doas comparison. AR, EG, SK, NM and MV provided critical feedback. All authors provided comments that helped shape the manuscript.

**Competing interests**

Andreas Richter and Vassilis Amiridis are members of the editorial board of the journal.

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



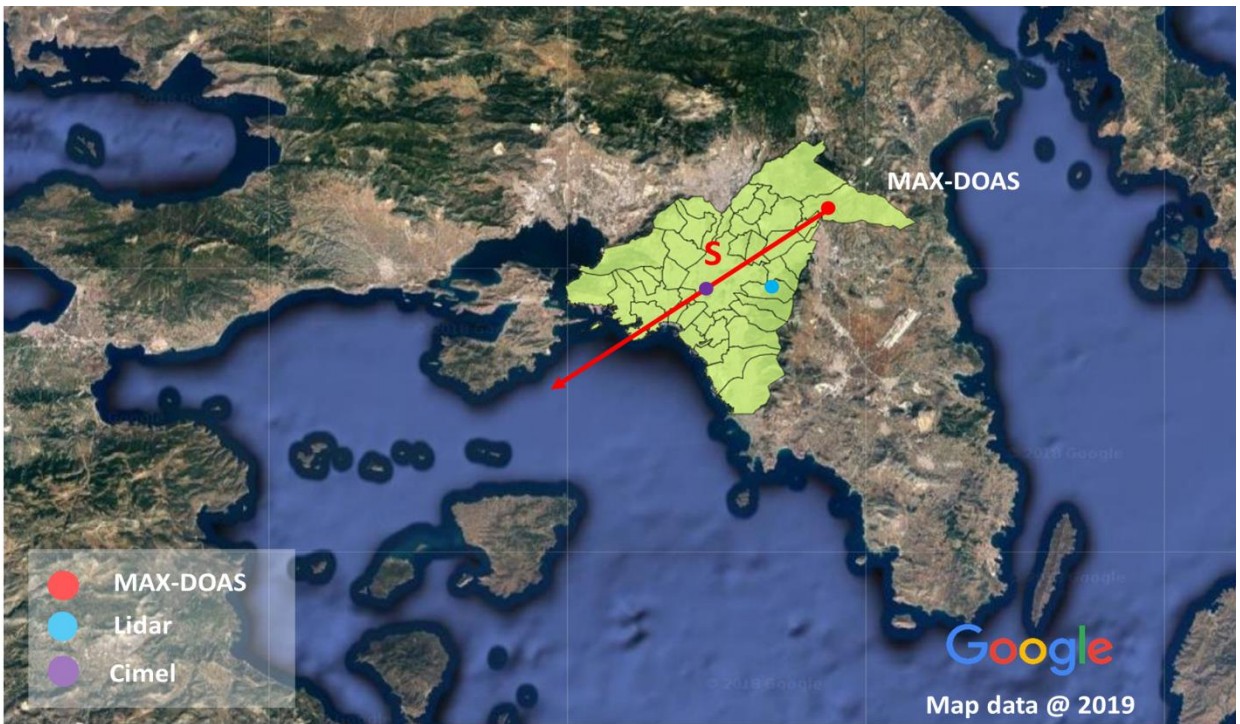

**Figure 1: Measurement sites and MAX-DOAS viewing direction (S). The distances between instruments are: MAX-DOAS - sun-photometer (CIMEL) 16 km and MAX-DOAS - lidar 13 km (© Google Maps).**

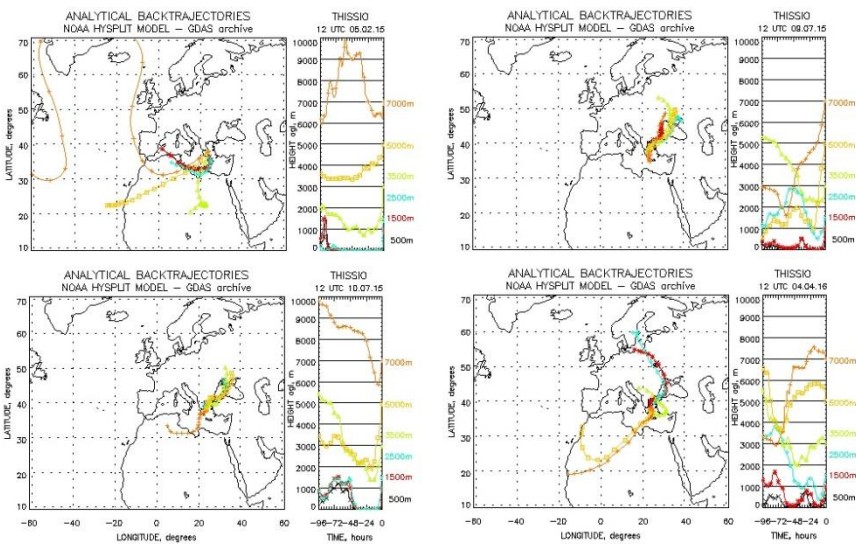

**Figure 2: Analytical backtrajectories for Athens as derived from NOAA-HYSPLIT model for the case studies (i) (top left panel), (ii) (top right panel, (iii) (bottom left panel) and (iv) (bottom right panel).**





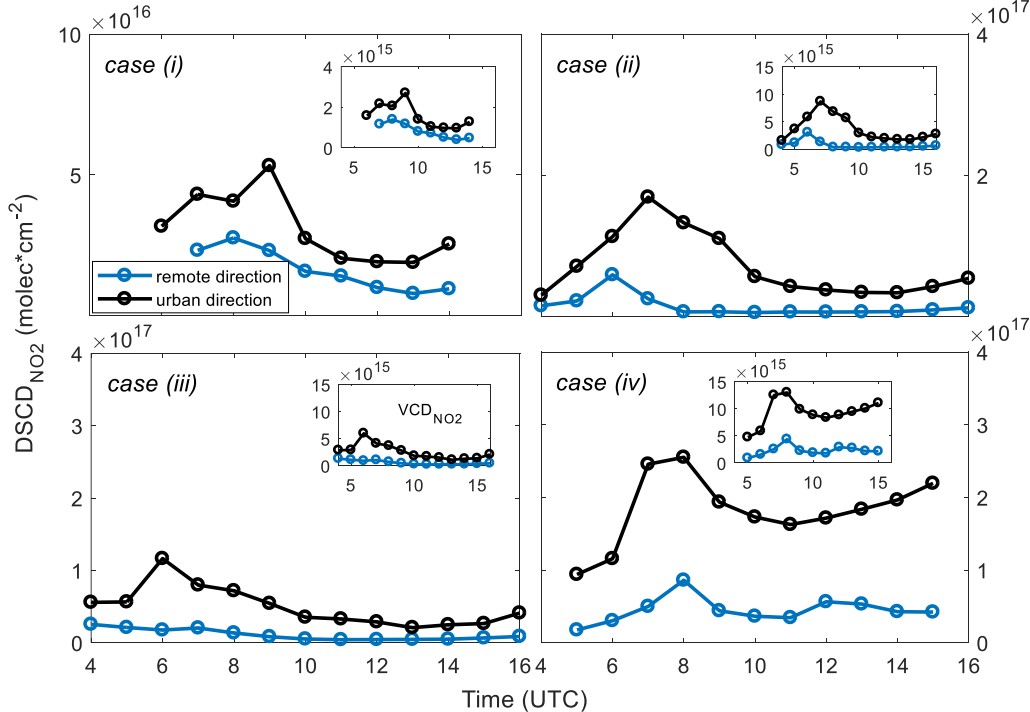

**Figure 3: Tropospheric SCD$_{NO2}$ retrievals (elevation angle +1º) from MAX-DOAS measurements for the four selected cases studies. The blue and the black curves correspond to the remote (W) and the urban (S) viewing direction, respectively. In the internal panels the corresponding tropospheric VCD$_{NO2}$ are also shown. Please consider the different scale used in case (i).**

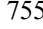

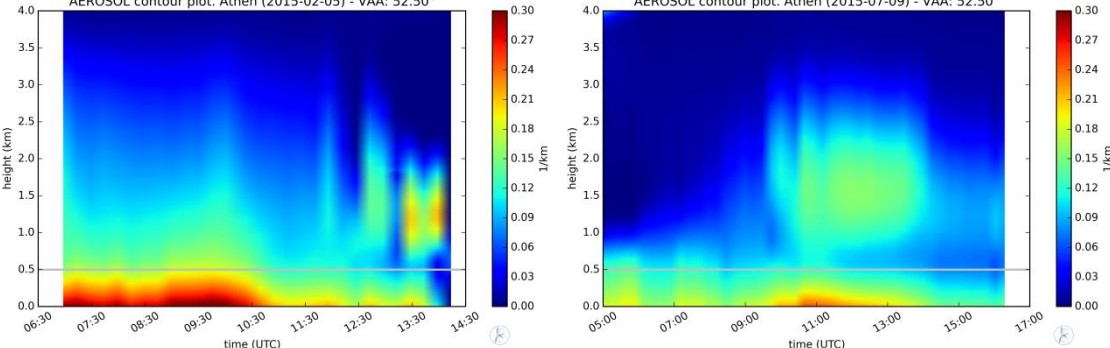



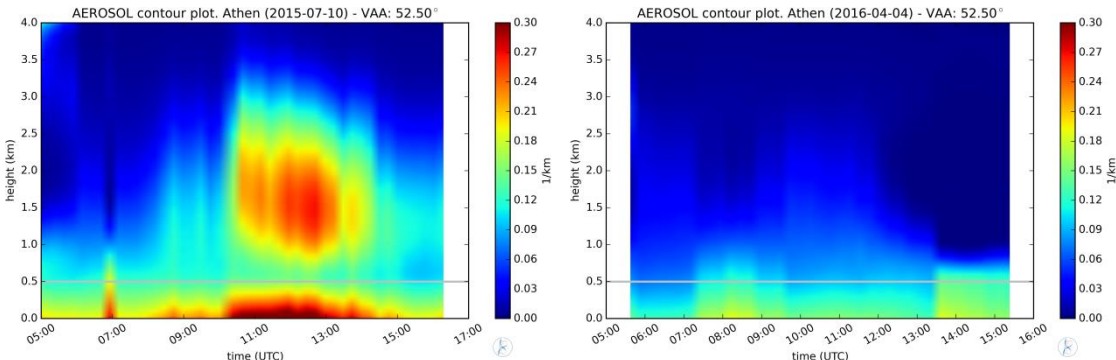

**Figure 4: MAX-DOAS retrieved aerosol extinction vertical distributions over the urban area (S) for case studies (i) (top left panel), (ii) (top right panel), (iii) (bottom left panel) and (iv) (bottom right panel). The grey line depicts the station's height.**


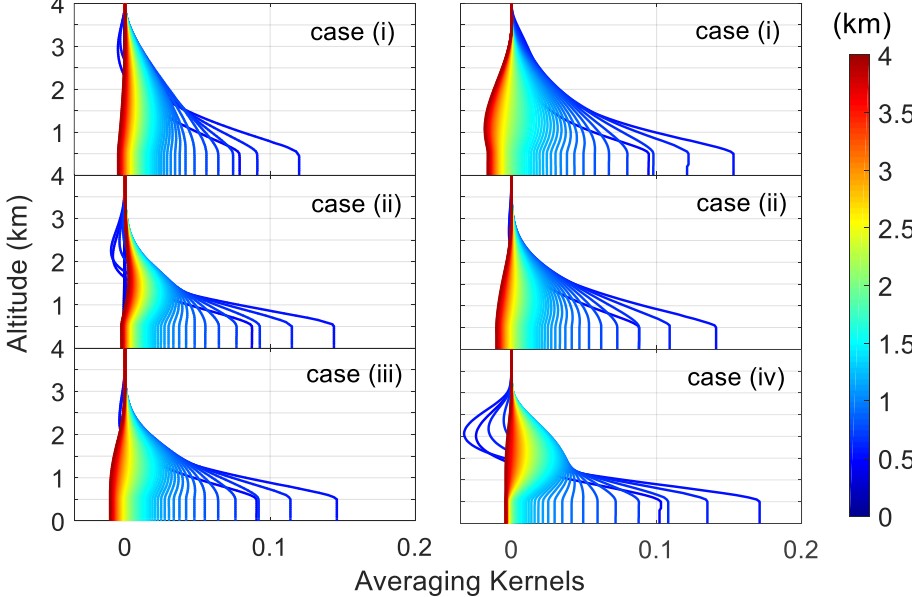

**Figure 5: Averaging kernels of the aerosol retrievals for the four case studies. For cases (i) and (ii), the left and right panel corresponds to the morning and afternoon profiles, respectively. The colour bar represents the height of the atmospheric layers.**









**Figure 6: Comparison of retrieved MAX-DOAS aerosol extinction profiles at 477 nm (multicoloured curves), lidar aerosol extinction coefficient vertical profile at 532 nm (black curve) and the corresponding degraded lidar profile (dashed black curve) for the selected case studies. The lidar profile used in each case is the average profile retrieved between the starting and the ending time of the MAX-DOAS retrievals and the light dashed black curves are the lidar-derived aerosol extinction uncertainty. The grey line depicts the elevation of MAX-DOAS station at 500 m.a.s.l.**


**Figure 7: AOD as derived from MAX-DOAS (black curve) and CIMEL at 440 nm and 500 nm (green and blue curve, respectively). The grey and the red square markers represent the Angström exponent derived from 400 and 870 nm and the lidar derived AOD, respectively. The dashed black curves represent the MAX-DOAS AOD uncertainties. The scatter plots between hourly AOD calculated from MAX-DOAS measurements and hourly AOD from CIMEL at 500nm are shown in the internal panels; the red points correspond to measurements until 16:00LT. Accordingly, $y_o$ is the linear regression equation with all the data points included and $y_1$ is the linear regression equation when the data points after 16:00LT have been excluded. The smaller green points are the raw data points. The vertical red dashed line separates the measurement data before and after 16:00LT.**





**Table 1: Instruments and data products used in the present study.**

| Instrument | Location | Institute | Products |
|---|---|---|---|
| MAX-DOAS | Penteli National Observatory of Athens (38.05°N, 23.86°E, 527 m a.s.l) | BREDOM network, Institute of Environmental Physics and Remote Sensing, University of Bremen | $SCD_{NO2}$, $VCD_{NO2}$, aerosol extinction profile, AOD |
| EOLE-LIDAR | Zografou (37.97°N, 23.79°E, 212 m a.s.l.) | National Technical University of Athens, Laser Remote Sensing Laboratory (NTUA-LRSU) | Aerosol backscatter profile, aerosol extinction profile, columnar AOD |
| CIMEL Sun-Sky Radiometer | Thissio (37.96°N, 23.72°E, 150 m a.s.l.) | National Observatory of Athens, Institute for Astronomy, Astrophysics Space Application & Remote Sensing (NOA-IAASARS) | AOD, Inversion data products (ssa, asymmetry factor, refractive index, phase function, size distribution) |

**Table 2. Settings used for the BOREAS retrieval. The mean daily value of each parameter (ω and g retrieved from AERONET) is mentioned for cases (i), (ii) and (iii). Next year's monthly mean of ω and g were used for case (iv), due to unavailable AERONET daily data around this date.**

|  | case (i) | case (ii) | case (iii) | case (iv) |
|---|---|---|---|---|
| **Surface albedo** | 0.15 | 0.15 | 0.15 | 0.15 |
| **Single scattering albedo (ω)** | 0.92 | 0.96 | 0.93 | 0.91 |
| **Asymmetry factor (g)** | 0.78 | 0.65 | 0.68 | 0.68 |
| **Tikhonov parameter** | 20 | 20 | 20 | 20 |


**Table 3. Quantitative performance statistics of MAX-DOAS aerosol extinction calculations (BOREAS algorithm) compared to lidar measurements.**

| Performance Measure | case (i)-mor | case (i)-aft | case (ii)-mor | case (ii)-aft | case (iii) | case (iv) |
|---|---|---|---|---|---|---|
| **r** | 0.98 | 0.84 | 0.86 | 0.95 | 0.97 | 0.86 |
| **median ratio (lidar/MAXDOAS)** | 0.92 | 0.45 | 0.84 | 0.57 | 0.91 | 1.80 |



| | | | | | |
|---|---|---|---|---|---|
| **RMSE (km$^{-1}$)** | 0.02 | 0.05 | 0.02 | 0.04 | 0.01 | 0.02 |
| **FGE** | 0.27 | 0.60 | 0.44 | 0.55 | 0.23 | 0.77 |

**Table 4. Quantitative performance statistics of MAX-DOAS AOD calculations (BOREAS algorithm) at 477 nm compared to CIMEL measurements at 500 nm .**

| Performance Measure | case (i) | case (ii) | case (iii) | case (iv) |
|---|---|---|---|---|
| **r** | 0.56 | 0.63 | 0.55 | -0.44 |
| **median ratio (CIMEL/MAXDOAS)** | 1.08 | 0.72 | 0.83 | 1.19 |
| **RMSE** | 0.03 | 0.09 | 0.13 | 0.05 |
| **FGE** | 0.09 | 0.35 | 0.32 | 0.27 |