# Peer review of "Retrieval and evaluation of tropospheric aerosol extinction profiles using MAX-DOAS measurements over Athens, Greece"

_Atmospheric Measurement Techniques, 2020_

## Referee Comment (RC1) · Anonymous Referee #1 · 26 May 2020

The paper by Gratsea et al. reports on MAX-DOAS measurements of aerosols at Athens, Greece. The MAX-DOAS were utilized to retrieve aerosol optical depths and vertical profiles of the aerosol extinction applying the BOREAS retrieval algorithm developed by the University of Bremen. The paper describes the potential and the application of a remote sensing technique to retrieve aerosol properties. Thus, the paper is relevant for the Atmospheric Measurement Techniques Journal. Although the study is based on a limited number of selected cases, provides a representative dataset for different atmospheric conditions of the under-study area. For the evaluation of the extinction coefficient retrievals, the aerosol extinction was compared with Lidar measurements and the aerosol optical depth with sky radiometer measurements, both in-

struments within a distance of 15km from the MAX-DOAS. Although the paper misses a comprehensive dataset, the 4 selected cases were modeled and compared with the measurement, showing promising results. The paper is well organized, and clear. I recommend the publication in AMT after revisions outlined in the following sections:

General comments

In order to avoid confusions, the same time zone should be used in the figures and throughout the manuscript. Please avoid the use of both LT and UTC. As the instruments used in the study are part of EU and International Infrastructure networks the use of the UTC is preferable.

The same unit format should be kept in the manuscript (e.g. m/s or ms-1).

It Is very difficult for the reader to follow the discussion and the Figures when only the case numbering is given. The discussion of the cases as well as the headers of the Figures should be based on the dates of each case or at least the dates and hours of the data should be given together with the case numbering in the plots.

Specific comments

Introduction

The authors nicely present the advantages of MAX-DOAS compared to established aerosol measurement techniques (e.g., simple and low cost instrumentation, the ability to perform long-term measurements also in remote areas, the ability to retrieve information on the vertical distribution of aerosol in contrast to sun photometers which only yield AOD) but the shortcomings and the limitations of the technique should also be mentioned in more detail in the introduction.

Section 2.2.2

One of the major points of the evaluation of the MAX-DOAS aerosol extinction retrievals is the comparison with the extinction lidar profiles. As the study makes use of daytime

lidar measurements an assumption of the lidar ratio is needed for the retrieval of the lidar extinction profile. In the manuscript the authors mentioned that the same lidar ratio is used for all cases. Did the authors check the lidar retrievals with e.g. comparison with AOD columnar observation from Cimel?

The uncertainty of the extinction lidar profiles should be discussed, estimated, and given in the manuscript.

In the L:177-178 the authors stand that the height independent extinction coefficient is representative for the aerosol load in the overlap region, is there any reference that supports this statement? The 1km of the overlap height range is still within the Planetary Boundary layer where an assumption like this could be accepted?

Section 2.3

Although already published elsewhere, the general approach and the main features of the algorithm and the optimal estimation method need to be described. For example, a definition of the box airmass factor is missing. What is the a priori aerosol Number concentration profile that is used for the BOREAS retrievals?

The authors should provide further information related to the extinction profiles uncertainties and possible biases in the evaluation with lidar kai sun-photometer retrievals due to the a priori selected values.

Table 2 provides information for the input parameters of the 4 selected cases, before the description of the selected cases in the manuscript. Table 2 could provide more generic information, or a rearrangement of the text is needed.

Section 3.1

A Table providing information (e.g date, atmospheric conditions, air masses), for the 4 selected cases may help the reader to have a better view of the differences and the similarities between the cases. Also, a table will facilitate the reader to follow the discussion which is referring in cases numbering and not in the dates of the cases.

L226: what kind of in situ meteorological observations provide information for cloud free conditions?

Figure 1: A closer map of the area with terrain could better highlight the special topography of the under-study region as well as the orientation and the elevation difference between the instruments. What is the green area in the Figure 1?

Section 3.2

Since the authors present a technique with main scope to deliver reliable results sensitivity studies are necessary. There is no information how much of the uncertainty of the retrieval is derived from the measurements and how much is from the a priori input. Furthermore, error bars in Figure 6 would help the reader to evaluate the retrieval.

Since AERONET measurement are used based on their availability either for the specific date or as a climatological mean value, the resulting uncertainty on the extinction profile should be further discussed and estimated.

Section 3.3

L275: Is the average of more than one extinction profiles, or the average of the lidar signal for the same time window as the MAX-DOAS retrievals? Please be specific.

L276: Please provide numerical estimation of the uncertainty in the extinction retrievals.

The authors should avoid general and non-specific comments, e.g L292: some discrepancies, L296: some performance statistics. Please rephrase.

L339: an aerosol layer of about 1.5 km deep. In which height?

Section 3.4

AOD calculations from Lidar

In which height range the lidar AOD have been calculated? It is limited to the first 4

km? is there any aerosol layer above 4km that may contribute to the AOD?

AOD evaluation with AERONET

It is possible the underestimation of the MAX-DOAS to be related to the fact that the AOD from sun photometer is referring to the total column of the atmosphere and the MAX-DOAS covers only the first 4km. Did authors examine the presence of aerosol layers above 4km (e.g lidar observations) for the selected cases? Is this the case for any of the 4 under study cases?

Additionally, the 370m height difference between the location of sunphotometer and MAX-DOAS could have contribution to the AOD differences. This point should further discussed in the manuscript.

Based on the altitude differences and keeping in mind the limitations of lidar to retrieve trustworthy extinction below the full overlap region and the fact that the MAX-DOAS provide profile up to 4km, a comparison of the AOD for the atmospheric layer between 1-4km could provide better conclusions. Is there a reason why this has not been done? The authors should consider to repeat the evaluation of the MAX-DOAS for different altitude ranges.

Section 4

There is a repetition in 2nd (L:455-463) and 5th (L485-490) paragraph. Please improve the text.

Line466: The authors should be more specific under which atmospheric conditions there is a better agreement. Statements like "in most cases" should be avoid. Please rephrase.

Technical corrections

Figure 1: Possible a map with terrain could better highlight the unique topography of the under-study region as well as the orientation and the elevation difference between

the instruments. What is the green area in the Figure 1?

Figure2: Case numbering should be added in the plots.

Figure 3: The Dates (and hours) should be given as a header in each plot together with the case (i-iv).

Figure 4: The case (i-iv) should be also given in the plots to facilitate the reader to follow. Please also provide the spatial and temporal analysis of the retrievals in the caption.

Figure 5: The Dates (and hours) should be given as a header in each plot together with the case (i-iv).

Figure 6: The Dates (and hours) should be given as a header in each plot.

Figure 7: The same axis (horizontal and vertical) should be used for each case. Please use the same x-axis (04-17 UTC) for all plots. Maybe a y-axis set at AOD=1.0 will make the plots less busy. Please keep the same format for each plot. The legend of top left plot seems incorrect (e.g lidar 520nm.) Please also mention the date for each case.

Table 2: Please correct . . .."Next year's monthly mean".

---

## Referee Comment (RC2) · Anonymous Referee #2 · 17 Jul 2020

The study "Retrieval and evaluation of tropospheric aerosol extinction profiles using MAX-DOAS measurements over Athens, Greece" presents aerosol profiles resulting from an inversion of MAX-DOAS measurements with the BOREAS algorithm. It generally matches the scope of AMT. However, there are major shortcomings in the methodology which require major clarifications, additional RTM calculations, and extended discussions. The manuscript should thus not be accepted for AMT unless major extensions and revisions are made. Thus, the current review only focusses on the retrieval part, as the presented results are likely going to change.

Retrieval shortcomings

[Figure]

**1. Observation geometry**

MAX-DOAS profile retrievals have been developed and refined in the last years and have been shown to yield valuable information on trace gas and aerosol profiles. However, the assumptions made in BOREAS (as well as other inversion schemes) put a MAX-DOAS instrument at the ground within flat terrain.

The situation in Athens seems to be quite different: the instrument is located at a hillside at ∼500m altitude. SCDs at negative and zero elevation angles have been measured, but are not included in the analysis. So I wonder how the near-surface extinction in Athens could be derived from an instrument on a hill looking upwards!? This aspect is not really explained and discussed quantitatively in the current manuscript.

I am not aware of a MAX-DOAS publication with similar viewing geometry. As the current study seems to be the first, it could be pioneering in this aspect. But it has to provide far more details, discussion and RTM calculations in order to interprete the resulting BOREAS profiles:

- what is the exact setup for the BOREAS retrieval? I assume the instrument was set to station altitude. But where is the surface in RTM calculations? At station level as well? At sea level? I assume that simulating 3D terrain is not easily possible, but the potential impact of terrain should at least be discussed.

- how large is the MAX-DOAS sensitivity for aerosols below station height? Fig. 5 indicates high sensitivity to altitudes below station level, but I really wonder where this should be coming from as the instrument is only looking upwards.

- what is the meaning of the aerosol contour plots down to sea level, and where is the information coming from?

- how has the AOD derived by BOAS to be interpreted? Is it really the full AOD for the city of Athens (where the MAX-DOAS is pointing at), or just the fractional AOD from station altitude upwards?

- from the contour plot, I would conclude that the largest fraction of the extinction profile is below 500 m. This could also be expected for urban pollution accumulating in a valley. But in order to interprete these results and the contour plots, it is essential to give evidence (by RTM calculations) that aerosol profiles can be actually derived down to the ground from an elevated intrument looking upwards. If the extinction below station altitude cannot be trusted, however, it should be discarded from Fig. 4. In this case, the comparisons should also only consider the fractional AOD from station altitude upwards. The case studies would than have to be revised completely.

2. Standard atmosphere

The profiles used for T and p affect the O4 VCD, which affects the aerosol profile inversion. Using a standard atmosphere is thus not appropriate. I recommend to repeat the analysis with more realistic T/p profiles for Athens. At least, the authors have to quantify the effect of using a standard atmosphere for winter vs. summer.

Minor comments:

Eq. 1: It might be worth mentioning that SCD_alpha is actually rather SCD_alpha-SCD_90 as well, as the zenith SCD has been used as reference in the DOAS analysis.

Page 8, line 234 and line 237: This information should already be given in section 2.2.1.

---

## Author Response (AR1)

**Response to reviewer #1**

We would like to thank the reviewer for reading this paper attentively and for suggesting both minor corrections and also corrections addressing main science issues. We respond to all his/her comments; the answers are given in blue.

General comments

**In order to avoid confusions, the same time zone should be used in the figures and throughout the manuscript. Please avoid the use of both LT and UTC. As the instruments used in the study are part of EU and International Infrastructure networks the use of the UTC is preferable.**

Done

**The same unit format should be kept in the manuscript (e.g. m/s or ms-1).**

Done

**It Is very difficult for the reader to follow the discussion and the Figures when only the case numbering is given. The discussion of the cases as well as the headers of the Figures should be based on the dates of each case or at least the dates and hours of the data should be given together with the case numbering in the plots.**

Dates and hours have been added in Figures 3, 4, 5, 6 and 7. Dates have also been added in the discussion section.

Introduction

**The authors nicely present the advantages of MAX-DOAS compared to established aerosol measurement techniques (e.g., simple and low cost instrumentation, the ability to perform long-term measurements also in remote areas, the ability to retrieve information on the vertical distribution of aerosol in contrast to sun photometers which only yield AOD) but the shortcomings and the limitations of the technique should also be mentioned in more detail in the introduction.**

We would like to thank the reviewer for this remark. The limitations have been added in lines 67-69 of the revised manuscript: i) MAX-DOAS' sensitivity at higher altitudes is low, ii) it provides profiles with much coarser vertical resolution compared to the lidar technique, iii) it performs only daylight measurements

Section 2.2.2

**One of the major points of the evaluation of the MAX-DOAS aerosol extinction retrievals is the comparison with the extinction lidar profiles. As the study makes use of daytime lidar measurements an assumption of the lidar ratio is needed for the retrieval of the lidar extinction profile. In the manuscript the authors mentioned that the same lidar ratio is used for all cases. Did the authors check the lidar retrievals with e.g. comparison with AOD columnar observation from Cimel?**

We thank the reviewer for this comment. We consider a lidar ratio input value of 50 ± 20 sr (as discussed in Section 2.2.2). This range covers the lidar ratio range for the pollution and dust cases presented in the manuscript (see Fig. 6 in Groß et al. (2013)). This range is also in accordance with columnar lidar ratio values (interpolated to 532 nm) obtained by AERONET for the cases of this study, which vary from 48.8 ± 7.5 sr to 59.9 ± 12.1 sr. The following text is now inserted in lines 172-174 of the revised manuscript:
"This range is realistic for pollution and dust cases presented herein (Groß et al., 2013) and it is also in accordance with columnar lidar ratio values (interpolated to 532 nm) obtained by AERONET for the cases of this study, which vary from 48.8 ± 7.5 sr to 59.9 ± 12.1 sr))."

Additionally, taking into account the constraints and the assumptions in the lidar ratio, the comparison between lidar-derived AOD and the corresponding AOD values retrieved by MAX-DOAS and AERONET is presented in Figure 7. The lidar AOD errors are now presented in a separate table (Table 7) in the revised manuscript.

**The uncertainty of the extinction lidar profiles should be discussed, estimated, and given in the manuscript.**

Thank you for raising this point, which was not clear.

Indeed, one of the greatest sources of uncertainty when solving the lidar equation for a common elastic backscatter lidar system is the assumption of a single lidar ratio value, which is considered constant for the entire atmosphere. In the initially submitted manuscript we mentioned that "…the overall uncertainty, including both statistical and systematic errors, on the retrieved $\beta_{aer}$ values, is of the order of 20–30% (e.g. Rocadenbosch et al., 2010). In this study, in order to account for the lidar ratio error assumption, we considered a lidar ratio input value of 50 ± 20 sr". This means that we solved the lidar equation with three different lidar ratio values as input (i.e. 30, 50 and 70 sr) in order to account for this wide variability.

Following the reviewer's suggestion, the following text has been added in lines 171-177 of the revised manuscript:

"In this study, the aerosol extinction profiles have been retrieved under the assumption of three typical lidar ratio values, 30, 50 and 70 (i.e. 50 ± 20 sr). This range is realistic and in accordance to columnar lidar ratio values (interpolated to

532 nm) obtained by AERONET for the cases of this study, which vary from 48.8 ± 7.5 sr to 59.9 ± 12.1 sr. As a result of this variability (i.e. 50 ± 20 sr), the uncertainties introduced to the aerosol extinction profiles vary from 10 - 40%; the higher uncertainties appear at the upper atmospheric layers, where the signal-to-noise ratio of the system decreases. The corresponding uncertainties for the lidar-derived AOD values of this assumption were estimated to be up to 11%."

Moreover, the uncertainties of the lidar aerosol profiles are shown in Figure 6 and the uncertainties of the lidar AOD are shown in Table 7 of the revised manuscript.

**In the L:177-178 the authors stand that the height independent extinction coefficient is representative for the aerosol load in the overlap region, is there any reference that supports this statement? The 1km of the overlap height range is still within the Planetary Boundary layer where an assumption like this could be accepted?**

We thank the reviewer for this comment. Indeed, we agree with the reviewer that the height-independent aerosol extinction coefficient value below 1 km a.s.l. may not be appropriate for the estimation of the aerosol load within the lowest atmosphere, which is mostly affected by the anthropogenic activity. However, this is the best assumption in order to partially account for the aerosol load at the lowest atmospheric layers. The text has been modified accordingly in lines 191-196 of the revised manuscript.

"Nevertheless, in order to calculate the AOD from the lidar profiles, the lowermost trustworthy value of the extinction coefficient was assumed constant down to the surface (height-independent). During daytime, the upper limit of the planetary boundary layer over Athens ranges between 1500 and 2100 m a.s.l. (Kokkalis et al. 2020), thus the minimum height of lidar profiles at 1000 m a.s.l. is well within the PBL. Our assumption of a well-mixed atmosphere below 1000 m a.s.l. - which means that a constant lidar ratio value is considered for this part of the atmosphere (Wandinger and Ansmann, 2002) - may lead to an underestimation of the AOD at the lowest troposphere, since the city is most probably a local source of the polluted particles. This underestimation and cannot be estimated because of the lidar overlap issue."

Section 2.3

**Although already published elsewhere, the general approach and the main features of the algorithm and the optimal estimation method need to be**

**described. For example, a definition of the box airmass factor is missing. What is the a priori aerosol Number concentration profile that is used for the BOREAS retrievals?**

The reviewer is right; we missed to describe important information. The information is now included in lines 208-213, 217-219 and 226-227 of the revised manuscript:

"The algorithm applies the optimal estimation technique for the retrieval of trace gas concentration profiles, while for our case - the aerosol retrievals - it uses an iterative Tikhonov regularization approach. The main concept of the algorithm for the aerosol retrievals is to minimize the difference between modeled and measured O4 slant optical depths by applying the iterative Tikhonov technique to varied aerosol extinction profiles. This method uses the difference of the slant optical depth from an a priori state in order to obtain information on the aerosol concentration that caused this difference through multiple iterations".

"The BAMF - in contrast to the total AMF - is a function of altitude describing the sensitivity of measurements to the profile at different atmospheric height layers. The aerosol weighting function matrices express the sensitivity of the o4 measurements to changes in the aerosol extinction coefficient profile. "
"...the a priori aerosol number concentration profile $N_\alpha(z)$ which is used as a starting point for the iterations... "

**The authors should provide further information related to the extinction profiles uncertainties and possible biases in the evaluation with lidar kai sun-photometer retrievals due to the a priori selected values.**

We would like to thank the reviewer for the suggestion to address the uncertainties of our calculations in section 2.3. We described the two different errors of our calculations and we calculated the uncertainty introduced to our calculations due to the a priori profile (this information is now included in Table 5 in the revised manuscript).

| Uncertainties (%) | case (i)-mor | case (i)-aft | case (ii)-mor | case (ii)-aft | case (iii) | case (iv) |
|---|---|---|---|---|---|---|
| smoothing error | 15.59 | 90.52 | 16.69 | 13.61 | 17.46 | 53.65 |
| noise error | 3.94 | 2.03 | 2.69 | 1.93 | 2.25 | 5.53 |

The following text has been added in lines 231-233 in section 2.3 of the revised manuscript: "The uncertainty associated with each retrieved profile is computed by the algorithm. It is the sum of the noise and smoothing error, which represent the impact of the measurements and of the a priori profile on the retrieved profile, respectively. These two errors have been calculated for each of our case studies separately and are presented in section 3.2."

**Table 2 provides information for the input parameters of the 4 selected cases, before the description of the selected cases in the manuscript. Table 2 could provide more generic information, or a rearrangement of the text is needed.**

The reference to Table 2 (renamed as Table 3 in the revised manuscript) has been moved to section 3.1.

Section 3.1

**A Table providing information (e.g date, atmospheric conditions, air masses), for the 4 selected cases may help the reader to have a better view of the differences and the similarities between the cases. Also, a table will facilitate the reader to follow the discussion which is referring in cases numbering and not in the dates of the cases.**

Thank you very much for this suggestion, which improves the appearance of the manuscript. Table 2 in the revised manuscript is now providing this information.

|                            | case (i)                                   | case (ii)                                   | case (iii)                       | case (iv)                   |
|----------------------------|--------------------------------------------|---------------------------------------------|----------------------------------|-----------------------------|
| **Date**                   | 5 Feb 15                                   | 9 Jul 15                                    | 10 Jul 15                        | 4 Apr 16                    |
| **Atmospheric conditions** | weak dust event, low pollution levels      | high pollution levels in the morning        | typical pollution levels         | high pollution levels       |
| **Air masses origin below 4 km** | S/SW                                 | N/NE                                        | N/NE                             | N/NE                        |

**L226: what kind of in situ meteorological observations provide information for cloud free conditions?**

These are empirical (line 255 of the revised manuscript) observations made by experienced staff of the National Observatory of Athens (NOA) and are registered to the NOA's official meteorological records.

**Figure 1: A closer map of the area with terrain could better highlight the special topography of the under-study region as well as the orientation and the elevation difference between the instruments. What is the green area in the Figure 1?**

Figure 1 has been updated in the revised manuscript so that the special topography of the city is more clear now.

Section 3.2

**Since the authors present a technique with main scope to deliver reliable results sensitivity studies are necessary. There is no information how much of the uncertainty of the retrieval is derived from the measurements and how much is from the a priori input.**

We would like to thank the reviewer for this comment. The a priori (smoothing) and measurement (noise) errors for each case are now presented in Table 5 in the revised manuscript.

**Furthermore, error bars in Figure 6 would help the reader to evaluate the retrieval.**

It was an omission from our side, thank you for pointing this out; the errors have been added in the plots of Fig. 6.

**Since AERONET measurement are used based on their availability either for the specific date or as a climatological mean value, the resulting uncertainty on the extinction profile should be further discussed and estimated.**

Thank you very much for pointing out the influence that the applied aerosol optical properties (SSA and asymmetry factor) have on the algorithm results. For addressing this, we carried out sensitivity tests with varying SSA and asymmetry factor. The values chosen for the sensitivity test, range between the minimum and maximum values for this season. In the following figure you may see the effect of different SSA and asymmetry factor values on the retrieved profile for case (iv) (which is the only case where a climatological mean was used) at 12:00UTC. The thick black line corresponds to the profile retrieved with the selected parameters for this study (g=68, ω=91). The variability due to asymmetry factor is small and the impact of SSA is negligible. This information is now added in lines 266-267 of the revised manuscript.

[Figure]

Section 3.3

**L275: Is the average of more than one extinction profiles, or the average of the lidar signal for the same time window as the MAX-DOAS retrievals? Please be specific.**

The reviewer is right; this point was not clear and is now rephrased in lines 311-313 of the revised manuscript: "The lidar profile presented in each figure is the result of the mean lidar signal, averaged between the starting and the ending time of the corresponding MAX-DOAS profiles."

**L276: Please provide numerical estimation of the uncertainty in the extinction retrievals.**

In L276 of the submitted manuscript (The uncertainty in the lidar extinction profiles increases substantially for altitudes below 1000 m.a.s.l...) we briefly refer to these uncertainties as an explanation for the reason why we use lidar data above that height in our studies. We believe that the numerical estimation of the uncertainty in the extinction retrievals from the lidar measurements below 1000 m a.s.l. is out of the scope of this study. We should emphasize though that this uncertainty does not affect the evaluation of MAX DOAS profiles with the corresponding lidar retrievals, since this is done for heights above 1000 m a.s.l.

**The authors should avoid general and non-specific comments, e.g L292: some discrepancies, L296: some performance statistics. Please rephrase.**

The non-specific comments have been rephrased in lines 315-317 and 319-321 of the revised manuscript:

"...hence the retrieved aerosol profiles from the two instruments correspond to different air masses and are not expected to fully agree, especially when the aerosol pollution is not horizontally homogeneous over the Athens basin. Thus, the comparison is mainly focused on a qualitative basis. "

"Comparison information is given in the form of performance statistics - correlation coefficient (r), median lidar/MAX-DOAS ratio, root mean square error (RMSE) and fractional gross error (FGE) – and is shown in Table 4. "

**L339: an aerosol layer of about 1.5 km deep. In which height?**

It has been rephrased in lines 364-365 of the revised manuscript:
"...an aerosol layer extending from the lower atmospheric layers up to 1.5 km height."

Section 3.4

AOD calculations from Lidar

**In which height range the lidar AOD have been calculated? It is limited to the first 4 km? is there any aerosol layer above 4km that may contribute to the AOD?**

The process followed for the estimation of the AOD from the lidar signal is summarized below:

(a) For the derivation of range-resolved aerosol optical properties, an aerosol-free reference height window has to be detected initially, where the normalized range-corrected lidar signal fits sufficiently the calculated attenuated molecular backscatter coefficient (Rayleigh-fit criterion; Freudenthaler et al., 2018). For this, the user provides the SCC input platform with an initial guess of that range (in our case 4-6 km based on visual inspection of the range-corrected lidar signal) and the corresponding algorithm fine tunes this guess,by applying different statistical tests to ensure that the shape of the measured signal corresponds to the shape of a Rayleigh signal (Mattis et al., 2016).

(b) The retrieval of the aerosol backscatter coefficient from the lidar signal starts from the identified reference height using the assumption of the lidar ratio value. The aerosol extinction coefficient is then calculated from the aerosol backscatter coefficient by multiplying it with the assumed lidar ratio.

(c) The columnar AOD is derived from the integration of the aerosol extinction profile from ground up to the identified reference height.

No significant aerosol load was observed above the reference height of 4 km a.s.l., in the free troposphere, based on our careful visual inspection of the range-corrected lidar signals. In addition, the AOD variability above the reference height lies within the 3rd decimal place, further ensuring that no significant aerosol layer contributes to the AOD.

Freudenthaler, V., Linné, H., Chaikovski, A., Rabus, D., and Groß,S.: EARLINET lidar quality assurance tools, Atmos. Meas. Tech.Discuss., https://doi.org/10.5194/amt-2017-395, in review, 2018

Mattis, I., D'Amico, G., Baars, H., Amodeo, A., Madonna, F., Iarlori, M.: EARLINET Single Calculus Chain – technical – Part 2: Calculation of optical products, Atmos. Meas. Tech., doi:10.5194/amt-9-3009-2016, 9, 3009–3029, 2016

AOD evaluation with AERONET

**It is possible the underestimation of the MAX-DOAS to be related to the fact that the AOD from sun photometer is referring to the total column of the atmosphere**

**and the MAX-DOAS covers only the first 4km. Did authors examine the presence of aerosol layers above 4km (e.g lidar observations) for the selected cases? Is this the case for any of the 4 under study cases?**

The reviewer is right, the following brief reference based on his/her comment has been made in the revised manuscript: "...the calculated AOD is limited up to 4 km, while the AOD from CIMEL refers to the total atmospheric column."
The reviewer's suggestion to examine the presence of aerosols at altitudes higher than 4 km could contribute to the observed underestimation by MAX-DOAS; yes, we examined the presence of free tropospheric aerosol layers above the identified reference height (~ 4 km a.s.l.) for all cases with visual inspection of the lidar signal, and no significant aerosol load was observed, as discussed in the previous comment. Although section 3.4 is focused on the comparison with the AOD from CIMEL, the AOD from lidar measurements (calculated by integrating the aerosol extinction coefficient from ground up to the identified reference height of 4 km a.s.l.) is also presented indicatively in the revised manuscript.

**Additionally, the 370m height difference between the location of sunphotometer and MAX-DOAS could have contribution to the AOD differences. This point should further discussed in the manuscript.**

The text has been rephrased accordingly in lines 464-469 in the revised version of the manuscript.
"Considering that i) the sun-photometer is located downtown (150 m a.sl.), at lower altitude than the MAX-DOAS (527 m a.s.l.) and thus more sensitive to aerosols in the lower troposphere and ii) the absence of real measurements from MAX-DOAS for altitudes below 500 m a.s.l., an underestimation of the contribution of the urban pollution to the retrieved by MAX-DOAS AOD would be expected. Nevertheless, the MAX-DOAS seems to detect well the typical urban aerosols in the boundary layer; the mean AOD difference (CIMEL minus MAX-DOAS) of all the measurements is 0.03 with standard deviation 0.08)."
The underestimation by MAX-DOAS that is observed in the afternoon is more probably related to MAX-DOAS' viewing geometry as explained in lines 426-429 in the originally submitted manuscript.

**Based on the altitude differences and keeping in mind the limitations of lidar to retrieve trustworthy extinction below the full overlap region and the fact that the MAX-DOAS provide profile up to 4km, a comparison of the AOD for the atmospheric layer between 1-4km could provide better conclusions. Is there a reason why this has not been done? The authors should consider to repeat the evaluation of the MAX-DOAS for different altitude ranges.**

We thank the reviewer very much for this remark. He/She is right, the comparison between lidar and MAX-DOAS AOD should be done for a common altitude (1-4 km), since the inclusion of the lowermost atmospheric layer to the AOD calculation adds a lot of uncertainty due to measurement restrictions. The reason for not including this comparison at first was that we would like to focus on the evaluation of the

retrieved MAX-DOAS AOD by comparing it with the CIMEL AOD, which is a well established method for AOD retrieval. Nevertheless, we calculated the AOD for 1-4 km altitude and is now presented in Table 7 of the revised manuscript.

| AOD (1-4 km) | case (i)-mor | case (i)-aft | case (ii)-mor | case (ii)-aft | case (iii) | case (iv) |
|---|---|---|---|---|---|---|
| lidar | $0.24 \pm 0.04$ | $0.21 \pm 0.03$ | $0.13 \pm 0.03$ | $0.19 \pm 0.03$ | $0.19 \pm 0.03$ | $0.09 \pm 0.01$ |
| MAX-DOAS | $0.16 \pm 0.03$ | $0.15 \pm 0.06$ | $0.18 \pm 0.04$ | $0.27 \pm 0.05$ | $0.19 \pm 0.04$ | $0.07 \pm 0.03$ |

Section 4
**There is a repetition in 2nd (L:455-463) and 5th (L485-490) paragraph. Please improve the text.**

The reviewer is right; there was a clear repetition, which has been corrected.

**Line466: The authors should be more specific under which atmospheric conditions there is a better agreement. Statements like "in most cases" should be avoid. Please rephrase.**

It has been rephrased (lines 503-505 in the revised manuscript):
"...there is good agreement in aerosol layer shape and aerosol extinction levels, except in cases of inhomogeneity at higher altitudes, characteristic of aerosol dust transport episodes. Very good correlation ($r > 0.90$) was found in all cases."

Technical corrections
**Figure 1: Possible a map with terrain could better highlight the unique topography of the under-study region as well as the orientation and the elevation difference between the instruments. What is the green area in the Figure 1?**

Figure 1 has been updated so that the special topography of the city is more clear now.

**Figure 2: Case numbering should be added in the plots.**

Unfortunately this cannot be done since the figures have been generated automatically by the HYSPLIT model. Nevertheless, the dates corresponding to each case have been added in the figure caption.

**Figure 3: The Dates (and hours) should be given as a header in each plot together with the case (i-iv).**

Dates have been added to Figure 3. It has been made more clear in the figure caption that the retrievals shown are the diurnal SC measurements.

**Figure 4: The case (i-iv) should be also given in the plots to facilitate the reader to follow. Please also provide the spatial and temporal analysis of the retrievals in the caption.**

Done.

**Figure 5: The Dates (and hours) should be given as a header in each plot together with the case (i-iv).**

Done.

**Figure 6: The Dates (and hours) should be given as a header in each plot.**

Done.

**Figure 7: The same axis (horizontal and vertical) should be used for each case. Please use the same x-axis (04-17 UTC) for all plots. Maybe a y-axis set at AOD=1.0 will make the plots less busy. Please keep the same format for each plot. The legend of top left plot seems incorrect (e.g lidar 520nm.) Please also mention the date for each case.**

We thank the reviewer for this comment which has improved the appearance of Fig. 7. The figures are now of the same format, with the same y-axes and the date mentioned along with the case number. The reason for keeping the different axes at the internal panels is the better presentation of the scatter plots; when same axes are used, the results are hard to read.

**Table 2: Please correct . . ..”Next year's monthly mean”.**

Rephrased in the caption of Table 2: "The mean monthly values of ω and g (provided from AERONET for April 2017) were used for case (iv), due to unavailable AERONET daily data around this date."

**Response to reviewer #2**

We would like to thank the reviewer for his/her constructive input and mainly for raising the important issue of the algorithm's performance for the retrievals below station's altitude. We respond to his/her comments; the answers are given in blue.

The study "Retrieval and evaluation of tropospheric aerosol extinction profiles using MAX-DOAS measurements over Athens, Greece" presents aerosol profiles resulting from an inversion of MAX-DOAS measurements with the BOREAS algorithm. It generally matches the scope of AMT. However, there are major shortcomings in the methodology which require major clarifications, additional RTM calculations, and extended discussions. The manuscript should thus not be accepted for AMT unless major extensions and revisions are made. Thus, the current review only focusses on the retrieval part, as the presented results are likely going to change.

Retrieval shortcomings

1. Observation geometry
MAX-DOAS profile retrievals have been developed and refined in the last years and have been shown to yield valuable information on trace gas and aerosol profiles. However, the assumptions made in BOREAS (as well as other inversion schemes) put a MAX-DOAS instrument at the ground within flat terrain.
The situation in Athens seems to be quite different: the instrument is located at a hillside at _500m altitude. SCDs at negative and zero elevation angles have been measured, but are not included in the analysis. So I wonder how the near-surface extinction in Athens could be derived from an instrument on a hill looking upwards!? This aspect is not really explained and discussed quantitatively in the current manuscript.

We would like to thank the reviewer for his critical remarks which helped us to realise a misinterpretation of our results. Indeed, the station's location is unusual and the retrieval of extinction values below station altitude needs further discussion. In principle, some light reflected on the surface and scattered in the atmosphere below the station altitude will be scattered also into upward pointing line of sights in particular for the lowest elevation angles. However, over dark surfaces this is a relatively small contribution to the total intensity and thus carries limited information on extinction in the lowest layers. In addition, the RTM SCIATRAN is a 1d model and cannot account for any effects related to the complex topography.

In response to the reviewer's comments, we have performed a series of sensitivity studies performing retrievals on synthetic data created using different vertical profiles of aerosol extinction. The results indicate that the extinction retrieved below station altitude is dominated by the a priori, scaled to the total retrieved AOD. In view of these disappointing results, we have revised the manuscript by removing all results below station altitude and changing the discussion accordingly.

**I am not aware of a MAX-DOAS publication with similar viewing geometry. As the current study seems to be the first, it could be pioneering in this aspect.**

MAX-DOAS Measurements from elevated stations have been reported in a number of publications, including Gomez et al., 2014, Schreier et al., 2016, Bognar et al., 2020, Ma et al., 2020, Wang et al., 2020). However, with the exception of Bognar et al. and Wang et al., they do not attempt a full profile retrieval.

Gomez, L., Navarro-Comas, M., Puentedura, O., Gonzalez, Y., Cuevas, E., and Gil-Ojeda, M.: Long-path averaged mixing ratios of O3 and NO2 in the free troposphere from mountain MAX- DOAS, Atmos. Meas. Tech., 7, 3373–3386, doi:10.5194/amt-7- 3373-2014, 2014

Schreier, S. F., Richter, A., Wittrock, F. and Burrows, J. P.: Estimates of free-tropospheric NO2 and HCHO mixing ratios derived from high-altitude mountain MAX-DOAS observations at midlatitudes and in the tropics, Atmos. Chem. Phys., 16(5), 2803–2817, doi:10.5194/acp-16-2803-2016, 2016.

Bognar, K., Zhao, X., Strong, K.,Chang, R. Y.-W., Frieß, U.,Hayes, P. L., et al. (2020).Measurements of tropospheribromine monoxide over four halogenactivation seasons in the Canadianhigh Arctic. Journal of GeophysicalResearch: Atmospheres, 125,e2020JD033015. https://doi.org/10.

Ma, J., Dörner, S., Donner, S., Jin, J., Cheng, S., Guo, J., Zhang, Z., Wang, J., Liu, P., Zhang, G., Pukite, J., Lampel, J., and Wagner, T.: MAX-DOAS measurements of NO2, SO2, HCHO, and BrO at the Mt. Waliguan WMO GAW global baseline station in the Tibetan Plateau, Atmos. Chem. Phys., 20, 6973–6990, https://doi.org/10.5194/acp-20-6973-2020, 2020.

Wang, Z., Chan, K. L., Heue, K.-P., Doicu, A., Wagner, T., Holla, R., and Wiegner, M.: A multi-axis differential optical absorption spectroscopy aerosol profile retrieval algorithm for high-altitude measurements: application to measurements at Schneefernerhaus (UFS), Germany, Atmos. Meas. Tech., 13, 1835–1866, https://doi.org/10.5194/amt-13-1835-2020, 2020.

**But it has to provide far more details, discussion and RTM calculations in order to interpret the resulting BOREAS profiles:**
**- what is the exact setup for the BOREAS retrieval? I assume the instrument was set to station altitude. But where is the surface in RTM calculations? At station level aswell? At sea level? I assume that simulating 3D terrain is not easily possible, but the potential impact of terrain should at least be discussed.**

Indeed, the instrument was set to station's altitude in the RTM calculations. The surface was set at sea level (this is now mentioned in lines 222-223 of the revised manuscript). As mentioned above, SCIATRAN is a 1d model, which does not allow modeling the effects of a variable topography.

**- how large is the MAX-DOAS sensitivity for aerosols below station height? Fig. 5 indicates high sensitivity to altitudes below station level, but I really wonder where this should be coming from as the instrument is only looking upwards.**

As discussed above, we have tested the sensitivity using synthetic data. As it turned out, the sensitivity is very low and the results are thus dominated by the a priori profile shape. This was not correctly reflected in the averaging kernels shown in Fig. 5 as for the atmospheric layers below station's height, the value for 500 m was shown. This has been corrected in the revised manuscript.

**- what is the meaning of the aerosol contour plots down to sea level, and where is the information coming from?**

As this information was mainly coming from the a priori, we have decided to remove it from Fig. 4 in the revised manuscript, which now presents only retrievals from instrument's height up to 4 km a.s.l.

**- how has the AOD derived by BOAS to be interpreted? Is it really the full AOD for the city of Athens (where the MAX-DOAS is pointing at), or just the fractional AOD from station altitude upwards?**

In the originally submitted manuscript, the AOD from BOREAS was the integration of the extinction values from the surface up to 4 km. In the revised paper, we present both the fractional AOD (1-4 km) in Table 7 of the revised manuscript and the AOD for the whole column by setting a constant value (equal to the retrieved value at 500 m) for the lower levels, where the retrievals are not trustworthy (Fig. 7).

**- from the contour plot, I would conclude that the largest fraction of the extinction profile is below 500 m. This could also be expected for urban pollution accumulating in a valley. But in order to interprete these results and the contour plots, it is essential to give evidence (by RTM calculations) that aerosol profiles can be actually derived down to the ground from an elevated intrument looking upwards. If the extinction below station altitude cannot be trusted, however, it should be discarded from Fig. 4. In this case, the comparisons should also only consider the fractional AOD from station altitude upwards. The case studies would than have to be revised completely.**

We agree with the reviewer that the MAX-DOAS retrievals of the extinction coefficient for heights below 500 m need to be optimized and that the measurements with negative elevation angles need to be included in the algorithm retrievals. However, for technical reasons, the measurements at negative elevation angles can currently not be included in the retrieval, even if good knowledge of thesurface albedo is assumed. As discussed above, we performed several RTM simulations with synthetic data in order to assess the sensitivity for aerosol layers below 500m (for which in our case using only upward observations, the information comes only from multiple scattering). However, the results were discouraging and we therefore decided to follow the reviewer's suggestion to exclude the retrievals below 500 m in the revised manuscript. The results excluding the retrievals of the lower layers are shown in the revised Figure 4. We are not attaching the results from the RTM calculations in this reply, but they are at your disposal if needed.

Regarding the AOD calculation, the missing values in the extinction coefficient profiles below 500 m are set to a constant value (equal to the retrieved value at 500 m). This assumes that the atmosphere is well-mixed below 500 m, which probably results in an underestimation of the calculated AOD in case of enhanced surface aerosol layer. In the revised manuscript we provide the new AOD calculations for

MAX-DOAS in Figure 7, along with their comparison with the sun-photometer AOD in Table 6.

2. Standard atmosphere

**The profiles used for T and p affect the O4 VCD, which affects the aerosol profile inversion. Using a standard atmosphere is thus not appropriate. I recommend to repeat the analysis with more realistic T/p profiles for Athens. At least, the authors have to quantify the effect of using a standard atmosphere for winter vs. summer.**

We would like to thank the reviewer for this important suggestion. We rerun all the retrievals using measured T/p profiles for Athens from the Atmospheric Science Radiosonde Archive of the University of Wyoming (http://weather.uwyo.edu/upperair/bufrraob.shtml) (lines 220-222). We provide all the profiles and statistics anew in the revised manuscript.

Minor comments:

**Eq. 1: It might be worth mentioning that SCD_alpha is actually rather SCD_alpha-SCD_90 as well, as the zenith SCD has been used as reference in the DOAS analysis.**

Thank you very much for pointing out this omission; we now made it more clear in the revised manuscript:

$$VCD = \frac{SCD_\alpha - SCD_{90^o}}{(AMF_\alpha - AMF_{90^o})}$$

**Page 8, line 234 and line 237: This information should already be given in section 2.2.1.**

Thank you for this remark, we agree and this information has been moved to section 2.2.1 (lines 116-118).

**Marked-up manuscript version**

[revised manuscript text omitted]

**Figure 7: AOD as derived from MAX-DOAS (black curve) and CIMEL at 440 nm and 500 nm (green and blue curve, respectively). The grey and the red square markers represent the Angström exponent derived from 400 and 870 nm and the lidar derived AOD, respectively. The dashed black curves represent the MAX-DOAS AOD uncertainties. The scatter plots between hourly AOD calculated from MAX-DOAS measurements and hourly AOD from CIMEL at 500nm are shown in the internal panels; the red points correspond to measurements after 13:00UTC.until 16:00LT. Accordingly, $y_o$ is the linear regression equation with all the data points included and $y_1$ is the linear regression equation when the data points after 13:00UTC16:00LT have been excluded. The smaller bluegreen points are the raw data points. The vertical red dashed line separates the measurement data before and after 13:00UTC.16:00LT.**

**Table 1: Instruments and data products used in the present study.**

| Instrument | Location | Institute | Products |
|---|---|---|---|
| MAX-DOAS | Penteli National Observatory of Athens (38.05°N, 23.86°E, 527 m a.s.l) | BREDOM network, Institute of Environmental Physics and Remote Sensing, University of Bremen | $SCD_{NO2}$, $VCD_{NO2}$, aerosol extinction profile, AOD |
| EOLE-LIDAR | Zografou (37.97°N, 23.79°E, 212 m a.s.l.) | National Technical University of Athens, Laser Remote Sensing Laboratory (NTUA-LRSU) | Aerosol backscatter profile, aerosol extinction profile, columnar AOD |

| | | | |
|---|---|---|---|
| CIMEL Sun-Sky Radiometer | Thissio (37.96°N, 23.72°E, 150 m a.s.l.) | National Observatory of Athens, Institute for Astronomy, Astrophysics Space Application & Remote Sensing (NOA-IAASARS) | AOD, Inversion data products (ssa, asymmetry factor, refractive index, phase function, size distribution) |

**Table 2. Information about the selected case studies.**

| | case (i) | case (ii) | case (iii) | case (iv) |
|---|---|---|---|---|
| **Date** | 5 Feb 15 | 9 Jul 15 | 10 Jul 15 | 4 Apr 16 |
| **Atmospheric conditions** | weak dust event, low pollution levels | high pollution levels in the morning | typical pollution levels | high pollution levels |
| **Air masses origin below 4 km** | S/SW | N/NE | N/NE | N/NE |

**Table 3.** Settings used for the BOREAS retrieval. The mean daily value of each parameter (ω and g retrieved from AERONET) is mentioned for cases (i), (ii) and (iii). The mean  monthly values  of ω and g (provided from AERONET for April 2017) were used for case (iv), due to unavailable AERONET daily data around this date.

| | case (i) | case (ii) | case (iii) | case (iv) |
|---|---|---|---|---|
| **Surface albedo** | 0.15 | 0.15 | 0.15 | 0.15 |
| **Single scattering albedo (ω)** | 0.92 | 0.96 | 0.93 | 0.91 |
| **Asymmetry factor (g)** | 0.78 | 0.65 | 0.68 | 0.68 |
| **Tikhonov parameter** | 20 | 20 | 20 | 20 |

**Table 4.** Quantitative performance statistics of MAX-DOAS aerosol extinction calculations (BOREAS algorithm) compared to lidar measurements.

| Performance Measure | case (i)-mor | case (i)-aft | case (ii)-mor | case (ii)-aft | case (iii) | case (iv) |
|---|---|---|---|---|---|---|
| r | 0.97 | 0.96 | 0.92 | 0.95 | 0.97 | 0.90 |
| median ratio (lidar/MAXDOAS) | 1.37 | 1.11 | 0.91 | 0.60 | 0.99 | 1.58 |
| RMSE (km$^{-1}$) | 0.03 | 0.03 | 0.02 | 0.04 | 0.01 | 0.02 |
| FGE | 0.31 | 0.80 | 0.37 | 0.54 | 0.20 | 0.61 |

**Table 5. The MAX-DOAS average smoothing and noise errors (%) for each case study.**

| Uncertainties (%) | case (i)-mor | case (i)-aft | case (ii)-mor | case (ii)-aft | case (iii) | case (iv) |
|---|---|---|---|---|---|---|
| smoothing error | 15.59 | 90.52 | 16.69 | 13.61 | 17.46 | 53.65 |
| noise error | 3.94 | 2.03 | 2.69 | 1.93 | 2.25 | 5.53 |

**Table 6.** Quantitative performance statistics of MAX-DOAS AOD calculations (BOREAS algorithm) at 477 nm compared to CIMEL measurements at 500 nm.

| Performance | case (i) | case (ii) | case (iii) | case (iv) |
|---|---|---|---|---|

| Measure | | | | |
|---|---|---|---|---|
| r | 0.47 | 0.67 | 0.53 | -0.42 |
| median ratio (CIMEL/MAXDOAS) | 1.22 | 0.85 | 0.95 | 1.37 |
| RMSE | 0.07 | 0.06 | 0.11 | 0.07 |
| FGE | 0.17 | 0.26 | 0.28 | 0.40 |

**Table 7. MAX-DOAS (477 nm) and lidar (532 nm) AOD calculations for the atmospheric layer 1-4 km.**

| AOD (1-4 km) | case (i)-mor | case (i)-aft | case (ii)-mor | case (ii)-aft | case (iii) | case (iv) |
|---|---|---|---|---|---|---|
| lidar | 0.24 ± 0.04 | 0.21 ± 0.03 | 0.13 ± 0.03 | 0.19 ± 0.03 | 0.19 ± 0.03 | 0.09 ± 0.01 |
| MAX-DOAS | 0.16 ± 0.03 | 0.15 ± 0.06 | 0.18 ± 0.04 | 0.27 ± 0.05 | 0.19 ± 0.04 | 0.07 ± 0.03 |